# Platform-directed allostery and quaternary structure dynamics of SAMHD1 catalysis

Oliver J. Acton[1,2,5], Devon Sheppard[1], Simone Kunzelmann ®[3], Sarah J. Caswell[1,5], Andrea Nans ®[3], Ailidh J. O. Burgess ®[1], Geoff Kelly[4], Elizabeth R. Morris ®[1,6], Peter B. Rosenthal ®[2] ✉ & Ian A. Taylor ®[1] ✉

SAMHD1 regulates cellular nucleotide homeostasis, controlling dNTP levels by catalysing their hydrolysis into 2'-deoxynucleosides and triphosphate. In differentiated CD4+ macrophage and resting T-cells SAMHD1 activity results in the inhibition of HIV-1 infection through a dNTP blockade. In cancer, SAMHD1 desensitizes cells to nucleoside-analogue chemotherapies. Here we employ time-resolved cryogenic-EM imaging and single-particle analysis to visualise assembly, allostery and catalysis by this multi-subunit enzyme. Our observations reveal how dynamic conformational changes in the SAMHD1 quaternary structure drive the catalytic cycle. We capture five states at high-resolution in a live catalytic reaction, revealing how allosteric activators support assembly of a stable SAMHD1 tetrameric core and how catalysis is driven by the opening and closing of active sites through pairwise coupling of active sites and order-disorder transitions in regulatory domains. This direct visualisation of enzyme catalysis dynamics within an allostery-stabilised platform sets a precedent for mechanistic studies into the regulation of multi-subunit enzymes.

Sterile alpha motif and HD-domain containing protein 1 (SAMHD1) is a dNTP triphosphohydrolase that catalyses the breakdown of dNTPs to constituent 2'-deoxynucleoside and triphosphate[1,2], and is therefore an important regulator of cellular nucleotide homeostasis in mammalian cells[3]. SAMHD1 is expressed in most tissues[4,5] and has anti-HIV-1 activity in terminally differentiated myeloid cells and resting T-cells[6–12] through suppression of reverse transcription by depletion of the dNTP pool[13–16]. In HIV-2 and related simian viruses, this restriction is overcome by the lentiviral accessory protein Vpx that targets human SAMHD1 for degradation by the proteasomal pathway[9,11,12] through interactions with the cullin-4 ligase substrate adaptor DCAF1[17–19].

SAMHD1 is also an interferon induced gene[4,20] and regulator of the innate immune response[21,22]. Germ line mutations in SAMHD1 are associated with early onset stroke[23] and the hereditary autoimmune disease Aicardi-Goutières Syndrome, which mimics congenital viral infection[22,24]. SAMHD1 mutations have also been identified as potential drivers of Chronic Lymphocytic Leukaemia (CLL)[25–27] and are present in hypermutated cancers[28]. Conversely, high SAMHD1 expression in Acute Myelogenous Leukaemia (AML) patients is associated with reduction in efficacy of the nucleoside-analogue anti-cancer drugs Cytarabine, Decitabine, Sapacitabine and Clofarabine[29–33] as a result of SAMHD1 hydrolysis of their triphosphorylated forms[34–36]. Therefore, modulation of SAMHD1 activity by drugs or potentially through targeting of SAMHD1 for proteasomal degradation by the lentiviral accessory protein Vpx has been proposed as a strategy for improving anti-cancer and anti-HIV therapies[29,34,37].

More recently, roles for SAMHD1 in the regulation of DNA repair at stalled replication forks and the suppression of the interferon response by cytosolic nucleic acids have been proposed[38,39].

[1]Macromolecular Structure Laboratory, The Francis Crick Institute, 1 Midland Road, London NW1 1AT, UK. [2]Structural Biology of Cells and Viruses Laboratory, The Francis Crick Institute, 1 Midland Road, London NW11AT, UK. [3]Structural Biology Science Technology Platform, The Francis Crick Institute, 1 Midland Road, London NW1 1AT, UK. [4]The Medical Research Council Biomedical NMR Centre, The Francis Crick Institute, 1 Midland Road, London NW1 1AT, UK. [5]Present address: AstraZeneca, The Discovery Centre, 1 Francis Crick Avenue, Cambridge CB2 0AA, UK. [6]Present address: Department of Biosciences, University of Durham, Durham DH1 3LE, UK. ✉e-mail: peter.rosenthal@crick.ac.uk; ian.taylor@crick.ac.uk

Moreover, nucleic acid binding by SAMHD1 has also been suggested as a regulatory mechanism in both its anti-viral[40] and DNA-repair functions[41].

Human SAMHD1 contains 626 residues comprising an N-terminal nuclear localisation signal[42] (residues 11–14), sterile alpha motif (SAM) domain[43,44] (residues 34–109), HD phosphohydrolase domain[45] (residues 111–599) and a C-terminal region (residues 600-626) that is targeted by lentiviral Vpx to recruit SAMHD1 to the proteasome for degradation[19,46]. SAMHD1 dNTP triphosphohydrolase catalytic activity requires assembly into homo-tetramers[47]. Sequences at the N- and C-termini of the HD domain stabilise inter-monomer protein-protein interactions and incorporate four pairs of allosteric, AL1 and AL2, nucleotide-binding sites. Nucleotide binding at these sites promote tetramer-assembly and regulate SAMHD1 through combined binding of G-based (AL1) and 2'-deoxynucleoside (AL2) triphosphates[2,48–51]. GTP is the physiological ligand for AL1, although dGTP can also coordinate this site[16,50]. AL2, is specific for a dNTP with the following $K_A$ order of preference: dATP > dGTP > TTP > dCTP[52–55] and a magnesium ion bridges the triphosphates of each GTP-dNTP pair in adjacent AL1 and AL2 binding sites[48,49]. SAMHD1 activity is also cell-cycle regulated by CyclinA2/CDK2 phosphorylation at Threonine 592 (Thr592)[56–58] that is proposed to downregulate the activity through effects on tetramer stability[16]. However, there is growing evidence from in vitro experiments that phosphorylated SAMHD1 can still maintain catalytic function if dNTP levels are high[16,58–60], suggesting that the regulation of SAMHD1 activity is not solely through phosphorylation. Removal of regulation through Thr592 dephosphorylation or mutation has been proposed to allow SAMHD1 to inhibit HIV-1 in cycling cells[56]. Although more recently this activity in cycling cells has been shown to require attenuation of the HIV-1 reverse transcriptase[61].

SAMHD1 catalytic activity is directed by metal-ions co-ordinated by conserved His and Asp residues in the HD domain active site[2,48,49]. More recently, structural and enzymatic inhibition studies employing α-β imido dNTPs (dNpNHpp) determined how the catalytic mechanism utilizes an active site, bi-metallic iron-magnesium centre to position a hydroxide nucleophile in-line and a conserved residue H215 to catalyse phosphoester bond hydrolysis of a substrate dNTP[51,62]. The importance of SAMHD1 as a key regulator of dNTP metabolism[3,63], its potential as a therapeutic target[29,31,34,37], its dysregulation in cancer[25,63] and intense interest in SAMHD1 allosteric regulation, assembly and catalysis[2,16,48,51,52,62] makes it a prime target to study the dynamics of enzyme catalysis in a complex multi-subunit system.

Here we employ cryo-EM to capture high-resolution snapshots of dNTP-directed SAMHD1 tetramer assembly, steady-state catalysis, and nucleotide-depletion. We show how allosteric activation generates the stable SAMHD1 tetrameric core, how dynamic tertiary and quaternary conformational changes involving pairs of SAMHD1 monomers facilitate substrate binding and product release and how the nucleotide-depleted tetramer remains in a quasi-stable state. Our data provide a direct visualisation of SAMHD1 catalysis-dynamics that are driven by order-disorder transitions of regulatory domains on an allostery-stabilised platform of catalytic domains. These observations of protein dynamics not only illuminate the mechanism of a specific biological machine but also set precedents for understanding the dynamics of catalysis by multi-subunit enzymes.

## Results

### SAMHD1 tetrameric states

In order to visualise SAMHD1 catalysis, we designed a standardised SAMHD1 dNTP hydrolysis reaction containing four dNTP substrates and a GTP allosteric activator that enabled the rate of reaction and the proportion of each substate and product to be quantified throughout the course of the reaction using $^1H$ NMR spectroscopy (Supplementary Fig. 1 and Supplementary Table 1). Under these conditions, up to 1200 s, ~85 % turnover of all dNTPs, the reaction approximates to steady-state conditions with some substrate preference in the rank order of $k_{cat}$ with dGTP > dCTP > TTP > dATP as observed in previous studies[51–55,62]. At later times the rates slow substantially as the reaction enters substrate depletion conditions and at 2400 s > 99% of substrate has been hydrolysed.

We extracted samples for cryogenic-EM (cEM) imaging (Supplementary Fig. 2 and Supplementary Table 2) at different time points within the reaction to examine which species were present in the evolving ($T = 0$ & $T = 10$ s), steady-state ($T = 300$ s; 5 min & $T = 600$ s; 10 min) and substrate depleted ($T = 2400$ s; 40 min & $T = 5400$ s; 90 min) regimes of the reaction. Extensive 2D and 3D classification combined with 3D particle variability analysis and re-classification (Supplementary Fig. 3) enabled us to identify a single dimeric species and in total, five states of SAMHD1 tetramer. The maps produced for these six species vary in resolution, largely dependent on the particle count at different time points but have allowed us to visualise the SAMHD1 dimer and to build atomic models for the tetramer structures with resolutions ranging from 3.7 to 2.8 Å (Table 1 and Supplementary Figs. 4, 5). The states differ in their quaternary conformation (Fig.1, Supplementary Movie 1 and Supplementary Table 3), their nucleotide and metal content (Supplementary Table 4) and in their abundance at different phases of the reaction (Fig. 1a and Supplementary Table 5). Prior to addition of dNTPs with only GTP present, ($T = 0$ s) only SAMHD1 dimers are present. At the first timepoint ($T = 10$ s), taken just after initiation by addition of dNTPs and immediately blotted on the grid, the reaction contains around 50% SAMHD1 dimers, accompanied by three forms of SAMHD1 tetramer (State-I, State-II, and State-III) (Fig. 1a–c). In samples taken from the reaction at steady-state, dimers are nearly absent and the State-I, State-II, and State-III SAMHD1 tetramers predominate (Fig. 1c). Additionally, two further forms of the SAMHD1 tetramer (State-IV and State-V) are also present, but with low abundance. By contrast, in the substrate-depleted stages ($T = 2400$ s and $T = 5400$ s) State-IV and -V then predominate (Fig. 1d) along with minor amounts of State-II and State-III that reduce in abundance from the $T = 2400$ s to $T = 5400$ s timepoints.

### SAMHD1 tetramer assembly

A fast-kinetic analysis of SAMHD1 catalysis reports a protein concentration dependent lag-phase of the rate in the early part of the reaction before establishment of a steady-state rate (Supplementary Fig. 6). Our cEM imaging of this evolving stage ($T = 10$ s) corresponding to the lag-phase at 1−2 μM SAMHD1 revealed a large proportion (50%) of SAMHD1 dimers not assembled into active tetramers, suggesting that at least one component of the lag is a slow dimer to tetramer assembly step. SAMHD1 tetramers, observed in crystal structures, comprise four monomers with predominantly helical secondary structure, each containing a catalytic and regulatory domain, the latter comprised of a C-terminal lobe and lobe-linker (Supplementary Fig. 7). They have D2 symmetry and are built from two dimer-interfaces, Dimer-1 and Dimer-2, that each have extensive interactions[2,48,51]. Dimer-1 that we assign as Monomer-A/Monomer-B and Monomer-C/Monomer-D (Fig. 1 and Supplementary Fig. 8), has an interface comprising the AL1 GTP-binding site, surrounding residues K116, N137, Q142, R145, R451 and a sidechain network made up from Y146, Y154, Y155, S161, N163, H321, T423, N425 and Y432 (Supplementary Fig. 8a, c). Dimer-2 assigned as Monomer-A/Monomer-C and Monomer-B/Monomer-D, has an interface mediated largely through hydrogen-bonding between N358, D361, H364, S368 and R371 on adjacent α13 helices (Supplementary Fig. 8b, d), the packing of the C-terminal lobe and lobe-linker regions (residues 455-599) against the 2-fold related adjacent active sites and the interaction of R333, R352, K354 and N358 with dNTPs bound at AL2. Our cEM data for the solution dimer produced a map at only low-resolution for the dimer particle, albeit with some evidence of SAM domain density that is absent from the high-

**Table 1 | Cryo-EM data collection, refinement and validation statistics**

| | Inhibitor EMD-18729 PDB 8QXJ | State-I EMD-18730 PDB 8QXK | State-II EMD-18731 PDB 8QXL | State-III EMD-18732 PDB 8QXM | State-IV EMD-18733 PDB 8QXN | State-V EMD-18734 PDB 8QXO |
|---|---|---|---|---|---|---|
| Data collection and processing | | | | | | |
| Magnification | 130,000 | 75,000 | 75,000 | 75,000 | 75,000 | 75,000 |
| Voltage (kV) | 300 | 300 | 300 | 300 | 300 | 300 |
| Electron exposure ($e^-/Å^2$) | 48.6 | 33.0 | 33.0 | 33.0 | 33.0 | 33.0 |
| Defocus range (μm) | −1.0 to −3.5 | −1.0 to −3.5 | −1.0 to −3.5 | −1.0 to −3.5 | −1.0 to −3.5 | −1.0 to −3.5 |
| Pixel size (Å) | 1.08 | 1.09 | 1.09 | 1.09 | 1.09 | 1.09 |
| Symmetry imposed | D2 | D2 | C2 | C2 | C2 | C2 |
| Initial particle stack | 2,064,842 | 1,011,713[a] | 1,011,713[a] | 1,011,713[a] | 1,220,084[b] | 1,220,084[b] |
| Consensus map particle stack | / | 305,021 | 305,021 | 305,021 | 221,953 | 221,953 |
| Final particle stack | 139,594 | 78,613 | 126,449 | 80,757 | 59,042 | 26,844 |
| Map resolution (0.143 FSC threshold) (Å) | 2.65 | 2.66 | 2.82 | 2.94 | 2.98 | 3.43 |
| Refinement | | | | | | |
| Initial Model (PDB code) | 6TX0 | 6TX0 | 6TX0 | 6TX0 | 6TX0 | 6TX0 |
| Model resolution (0.5 FSC threshold) (Å) | 2.80 | 2.77 | 2.99 | 3.03 | 3.35 | 3.72 |
| Map sharpening B-factor | −117.4 | −94.6 | −103.0 | −97.5 | −123.0 | −121.9 |
| *Model composition* | | | | | | |
| Nonhydrogen atoms | 15953 | 15323 | 15060 | 14658 | 14461 | 14291 |
| Protein atoms | 15355 | 14947 | 14684 | 14340 | 14145 | 14041 |
| Ligand molecules | DAP, 8<br>GTP, 4<br>Mg, 12<br>Fe, 4 | dCTP, 4<br>dATP, 4<br>GTP, 4<br>Mg, 12<br>Fe, 4 | dC, 2<br>$P_3O_{10}H_4$, 2<br>dCTP, 2<br>dATP, 4<br>GTP, 4<br>Mg, 10<br>Fe, 4 | dCTP, 2<br>dATP, 4<br>GTP, 4<br>Mg 10<br>Fe, 4 | dCTP, 2<br>dATP, 4<br>GTP, 4<br>Mg, 8<br>Fe, 4 | dCTP, 2<br>dATP, 2<br>GTP, 4<br>Mg, 2<br>Fe, 4 |
| *Q-scores* | | | | | | |
| Protein | | | | | | |
| backbone | 0.78 | 0.78 | 0.70 | 0.72 | 0.63 | 0.58 |
| sidechains | 0.73 | 0.74 | 0.64 | 0.70 | 0.54 | 0.51 |
| Ligands | 0.86 | 0.84 | 0.67 | 0.80 | 0.51 | 0.62 |
| *R.M.S. deviations* | | | | | | |
| Bond lengths (Å) | 0.006 | 0.003 | 0.002 | 0.002 | 0.002 | 0.002 |
| Bond angles (°) | 0.995 | 0.587 | 0.494 | 0.528 | 0.532 | 0.541 |
| *Validation* | | | | | | |
| CC (main chain) | 0.88 | 0.83 | 0.81 | 0.84 | 0.75 | 0.69 |
| CC (side chain) | 0.85 | 0.80 | 0.78 | 0.82 | 0.71 | 0.67 |
| Molprobity Score | 1.36 | 1.22 | 1.37 | 1.34 | 1.46 | 1.43 |
| Clash score | 3.10 | 3.50 | 3.15 | 2.75 | 4.05 | 4.84 |
| Poor rotamers (%) | 0.12 | 0.00 | 0.06 | 0.00 | 0.07 | 0.27 |
| *Ramachandran* | | | | | | |
| Favoured (%) | 96.16 | 97.63 | 96.08 | 95.91 | 96.03 | 96.93 |
| Allowed (%) | 3.58 | 2.37 | 3.92 | 4.09 | 3.97 | 2.95 |
| Disallowed (%) | 0.26 | 0.00 | 0.00 | 0.00 | 0.00 | 0.12 |
| CaBLAM outliers (%) | 2.4 | 1.8 | 2.1 | 2.0 | 2.3 | 1.9 |

[a]Initial particle stack taken from a $T = 600$ s timepoint;
[b]Initial particle stack taken from a $T = 2400$ s timepoint.

resolution tetramer maps (Supplementary Fig. 9). Nevertheless, it is apparent that only a Dimer-1 model satisfies this density (Supplementary Fig. 9a, b). This observation supports the notion that Dimer-1, promoted by the presence of GTP, is prevalent as the solution dimer and provides the intermediate building block for tetramer assembly. By contrast, Dimer-2 appears to be only stable within the context of the assembled tetramer (Supplementary Fig. 9c).

## The steady state

At times subsequent to the evolving reaction, SAMHD1 tetrameric States I, II and III predominate. The State-I tetramer (Fig. 1b, c) has D2 symmetry with three perpendicular 2-fold axis that relate the Dimer-1 and Dimer-2 pairs. It closely resembles SAMHD1-substrate and inhibitor-complex crystal structures[48,51] as well as the cEM structure of a SAMHD1-GTP-dApNHpp inhibitor complex (Table 1 and

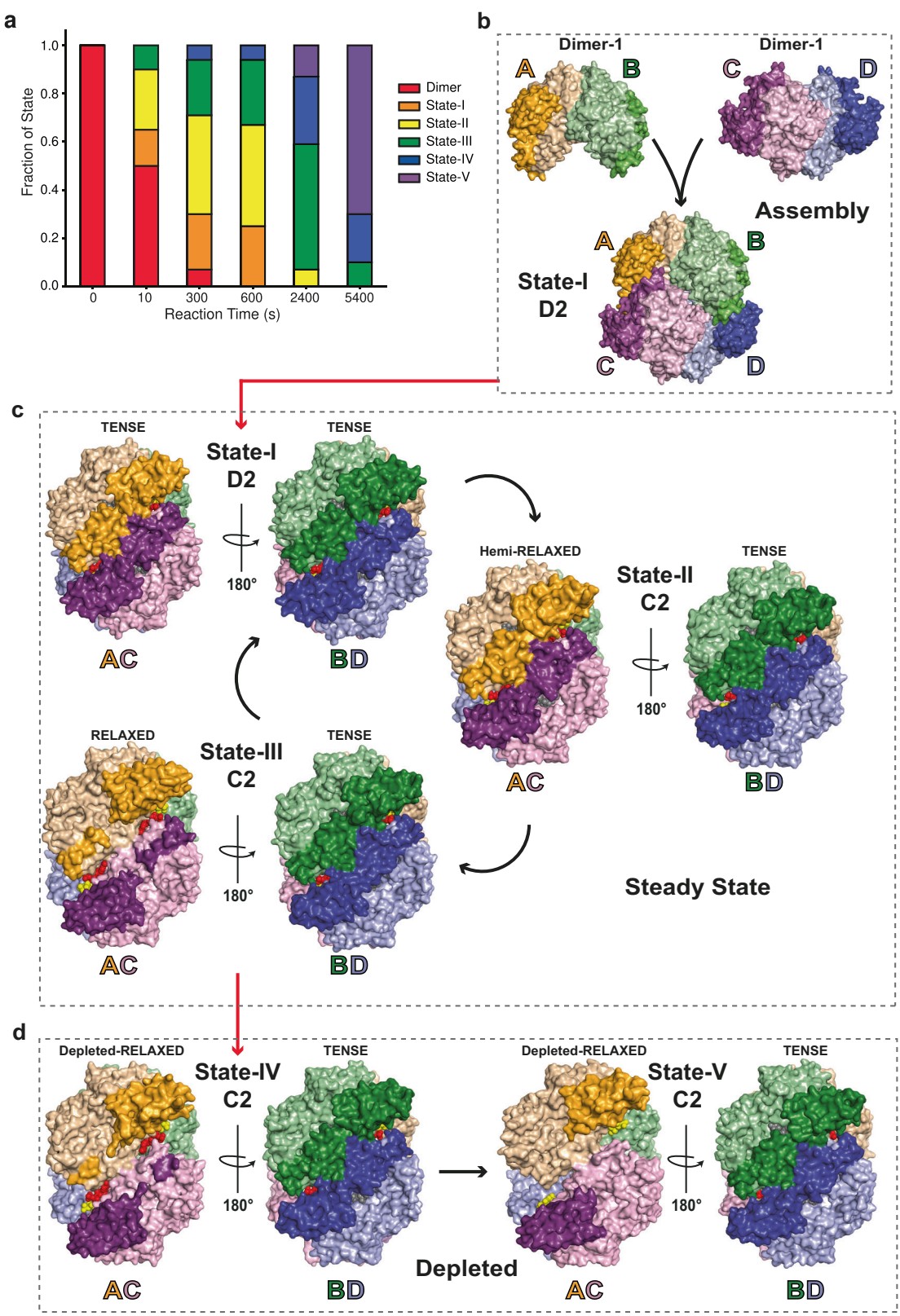

Supplementary Fig. 10). The map contains electron density for nucleotide substrates at the four active sites as well as GTP-dNTP pairs in all four AL1 - AL2 allosteric sites (Fig. 2). As these data were collected from a four-dNTP reaction, based on previously identified selectivity preferences[51–55,62], in our models we have placed dCTP at the active site and dATP at AL2 (Fig. 2a) although it is likely the active sites have mixed dNTP-base occupancy. Accompanying State-I D2

tetramers, and predominating in the later steady state timepoints, are two further tetramers, State-II and State-III that have only C2 symmetry (Fig. 1c). It is the cycling between State-I, State-II, and State-III that we propose represents the steady-state of SAMHD1 catalysis. State-II and State-III tetramers comprise two copies of an invariant Dimer-1, Monomer-A/Monomer-B and Monomer-C/Monomer-D, that further assemble into tetramers through Dimer-2

**Fig. 1 | cEM structures of the SAMHD1 catalytic cycle. a** Evolution of SAMHD1 states. Fraction of SAMHD1 dimer and tetramer states I−V, observed in cEM imaging of hydrolysis reactions is plotted against reaction time. Data from Supplementary Table 5. **b** Tetramer assembly. On addition of GTP and dNTPs two copies of SAMHD1 Dimer-1 associate to form the catalytically competent State-I D2 tetramer. Dimer from 6TX0 fitted into our low-resolution cEM map, Supplementary Fig. 9. Catalytic and regulatory lobe-linker domains of the four monomers coloured wheat and orange (Monomer-A), pale green and green (Monomer-B), pink and magenta (Monomer-C), pale blue and blue (Monomer-D). **c** The steady state, cycling between States-I to -III indicated by curved arrows. Tetramers are shown in surface, in two 180°-rotated views perpendicular to the AC and BD Dimer-2 two-fold axes. Catalytic and regulatory lobe-linker domains coloured as in **b**, active site nucleotides in grey, allosteric site nucleotides in red and yellow. State-I D2 tetramer has equivalent tense Dimer-2 AC and BD pairs, with fully ordered regulatory domain lobe-linkers. All active sites are substrate-loaded, and all allosteric sites are occupied. The State-II

C2 tetramer, Dimer-2 pairs are non-equivalent. The AC Dimer-2 is hemi-relaxed, with less ordered lobe-linkers. The BD Dimer-2 maintains the tense state. AC active sites contain products, BD active sites contain substrate, all allosteric sites are occupied. The State-III C2 tetramer, AC Dimer-2 is further relaxed with more disordered C-terminal lobe- linkers, active sites are exposed and empty. BD Dimer-2 retains the tense state with substrate-loaded active sites, all allosteric sites are occupied. **d** The depletion phase, States-IV and -V predominate. The State-IV C2 tetramer. AC Dimer-2 exhibits further disorder of lobe-linkers with accompanying empty active sites. BD Dimer-2 maintains the tense state with substrate-loaded active sites. All allosteric sites are occupied. The State-V C2 tetramer. AC Dimer-2 has entirely disordered C-terminal lobe-linkers and exposed empty active sites. BD Dimer-2 retains the tense state with substrate-loaded active sites. Allosteric nucleotides have been released from the AC Dimer-2 but are retained at BD allosteric sites.

interfaces between Monomer-B and Monomer-D and between Monomer-A and Monomer-C. However, whilst in the State-I tetramer these Dimer-2 interfaces are equivalent, in State-II and -III although they resemble the State-I Dimer-2 in that they are mediated through helix α13 interactions, on one side of the tetramer the Dimer-2 interface has substantially reduced packing of the C-terminal lobe-linker regions (residues 507-545). This asymmetry gives rise to C2 tetramers with a compact, tense side that we assign to the Monomer-B/Monomer-D interface and a relaxed or hemi-relaxed side assigned to the Monomer-A/Monomer-C interface (Fig. 1c and Supplementary Movie 1). Further examination of the nucleotide content reveals that on the tense side, in both State-II and State-III tetramers, the active sites contain substrate dNTPs, and the allosteric sites are loaded with GTP-dNTP pairs (Fig. 2b, c). On the relaxed side, the shifting of the C-terminal lobes partially exposes the active and allosteric sites (Fig. 1c and Supplementary Fig. 11) and, whilst the allosteric sites are fully occupied in both State II and State III the active sites contain either only weak density (State II) that we interpret as reaction products or are completely empty (State III) (Fig. 2b, c and Supplementary Fig. 11).

### Nucleotide depletion

In the substrate depleted phase of the reaction $T = 2400$ and $T = 5400$ s, two further C2 tetramer states now predominate, State IV and State V (Fig. 1d). State IV has a similar conformation to State III, with an active site-loaded tense side and nucleotide loss from the relaxed-side active sites (Fig. 2d). However, greater disorder in the C-terminal lobe linkers and an increased roll of the C-terminal lobe further exposes the active sites on the relaxed side (Fig. 1d). State V has even further disorder in the 507−545 lobe-linker and along with loss of active site deoxynucleotides, now additionally has loss of AL2 nucleotides on the relaxed side of the tetramer (Fig. 2e). Given our NMR data show that at these time points the reaction is greater than 99% complete (Supplementary Fig. 1c and Supplementary Table 5), it is likely that States-IV and -V represent non-actively catalysing tetramers that in the absence of nascent substrate are able to persist in a quasi-stable state.

### Active site configuration throughout the catalytic cycle

In the steady-state and substrate depletion phases of the reaction, the large-scale dynamic changes in the SAMHD1 quaternary structure are accompanied by concerted side chain movements within the active site. Inspection of the State-I active sites reveal density for nucleotides and metal ions bound (Figs. 2 and 3) with a configuration consistent with that observed in the high-resolution crystal structures of SAMHD1-dNPNHPP inhibitor complexes[51] and that of our cEM SAMHD1-dApNHpp structure (Supplementary Fig. 10). Therefore, we have assigned the active-site metals Fe, Mg2 and Mg3 in the State-I structure (Fig. 2a) based upon these previous extensive

characterisations of SAMHD1 metal ion composition[16,51,62]. Specifically, the phosphates of the bound nucleotide, engage the three active site metal ions Fe, Mg2 and Mg3, as well as making interactions with active site residues R164, H215, H233, K312, Y315 and R366 (Fig. 3a and Supplementary Fig. 12). Additionally, the nucleotide base is sandwiched in π-stacking interactions with H215 and Y374 and makes a hydrogen bonding interaction with the Q375 sidechain. In State-II, the active sites in the tense side of the tetramer maintain the same nucleotide configuration as in State-I. By contrast, on the relaxed side there is discontinuous and weaker density for the active-site contents and surrounding side chains, particularly Y315, that have undergone changes in configuration (Figs. 2b and 3c). Although there is a degree of disorder, placement of nucleoside and triphosphate products locates the discontinuity in the density between the α-phosphate of the triphosphate and 5′-hydroxl of the deoxyribose sugar, giving us confidence to model this relaxed side of the State-II tetramer as a product complex. Other notable features of State-II are the loss of the Mg2 ion that co-ordinates the β and γ phosphates of substrate dNTPs, and that the products have now been partially released from the active site. This product release is facilitated by a lever movement of the tyrosyl side chain of Y315 that in state-I interacts with the sugar and γ-phosphate of the substrate but is now rotated by 90° to occupy a position between the product α-phosphate and the Fe-ion at the HD-motif. This has the effect of displacing the triphosphate product to a distance of 4.5 to 5 Å from the Fe metal ion and the R164 sidechain, that in State-I are tightly co-ordinated to the substrate non-bridging α-phosphate oxygens.

In State-III, the active sites of the tense side of the tetramer also maintain the same nucleotide configuration as in State-I and State-II. However, the active sites on the relaxed side are now completely empty with no substrate or products bound (Figs. 2c and 3d). Moreover, as well as the loss of the Mg2 metal ion and Y315 movement seen in State-II, in State-III there is now also further rearrangement of the active site sidechains that make interactions with the deoxynucleotide bound in State I. Specifically, the R366 and Q375 side chains are rotated by about 90° away from their engaged positions and the aromatic ring of Y374 is shifted into the position that was occupied by the nucleotide base in State I (Fig. 3b−d and Supplementary Fig. 12c). In the depleted phase, the quasi-stable State-IV and -V tetramers retain empty active sites on the relaxed side and with further loss of Mg3 in State-V (Figs. 2d, e and 3e, f).

Based on our observations of active site residue motion and substrate interactions, to test our structural model of SAMHD1 catalysis we made mutations at three classes of active site residue to assess the impact on GTP/dATP hydrolysis measured by $^1$H NMR, and on GTP/dApNHpp induced tetramerisation measured by SEC-MALLS. Class-A constitutes residues critical for substrate binding, typified by R164, Class-B residues are required directly for catalytic chemistry, H215 and Class-C (R366 and Q375) are positioned to respond to substrate binding

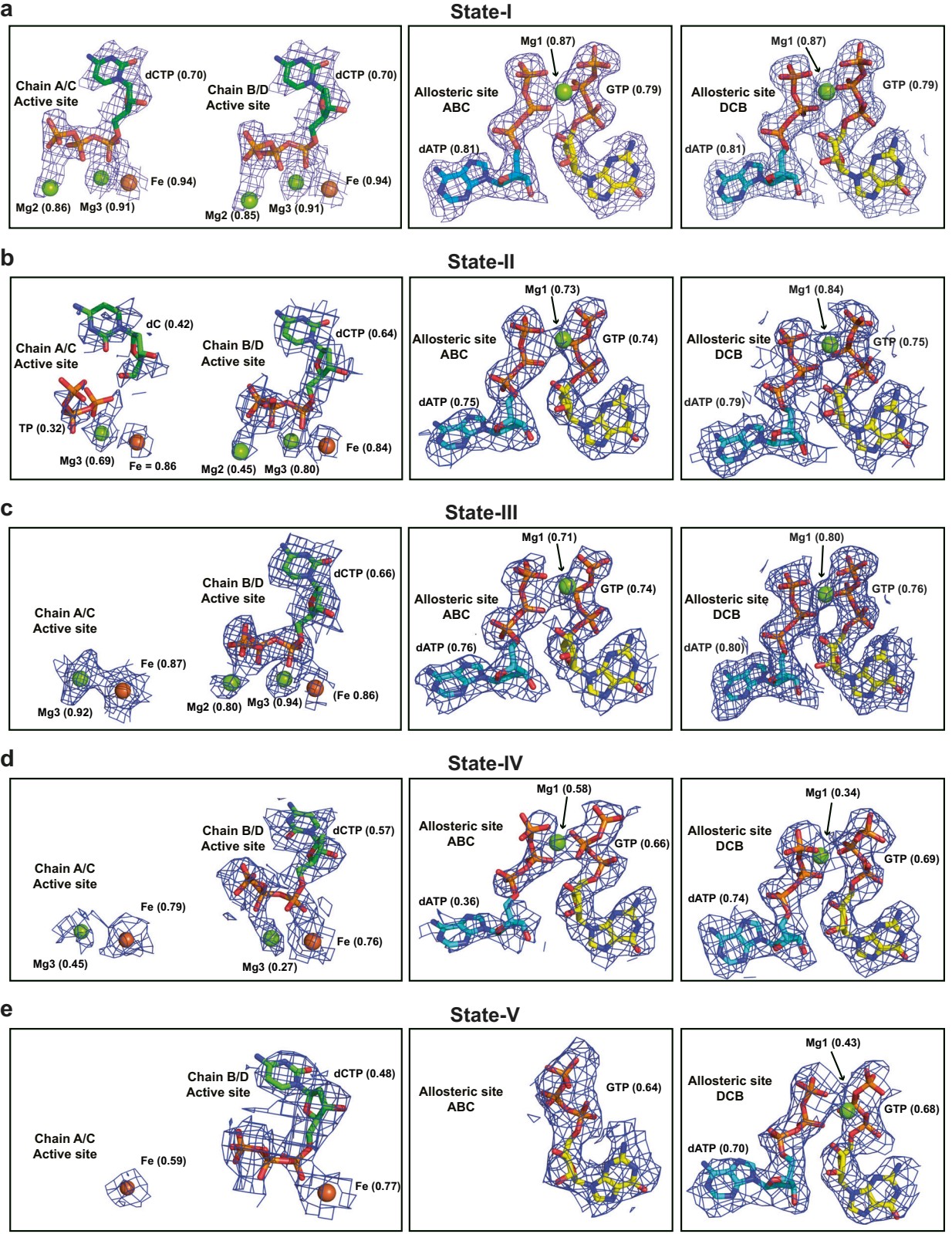

**Fig. 2 | Density for metal ions and nucleotides at active and allosteric sites in States -I to -V.** Electron density for bound nucleotides at Monomer-A and Monomer-B active sites and allosteric sites ABC and DCB in SAMHD1 cEM structures. **a** State-I (**b**) State-II (**c**) State-III (**d**) State-IV (**e**) State-V. In State-I, all four A, B, C and D active sites are identical, dictated by the D2 symmetry. In States-II–IV, Chain A relates to C and B to D through the C2 symmetry. In each panel nucleotides are shown in stick representation. Metal ions are shown as spheres, Fe (brown) Mg (green). cEM maps are shown as a blue mesh, contoured at 7 σ with map-model validation Q scores for nucleotides and ions displayed adjacently.

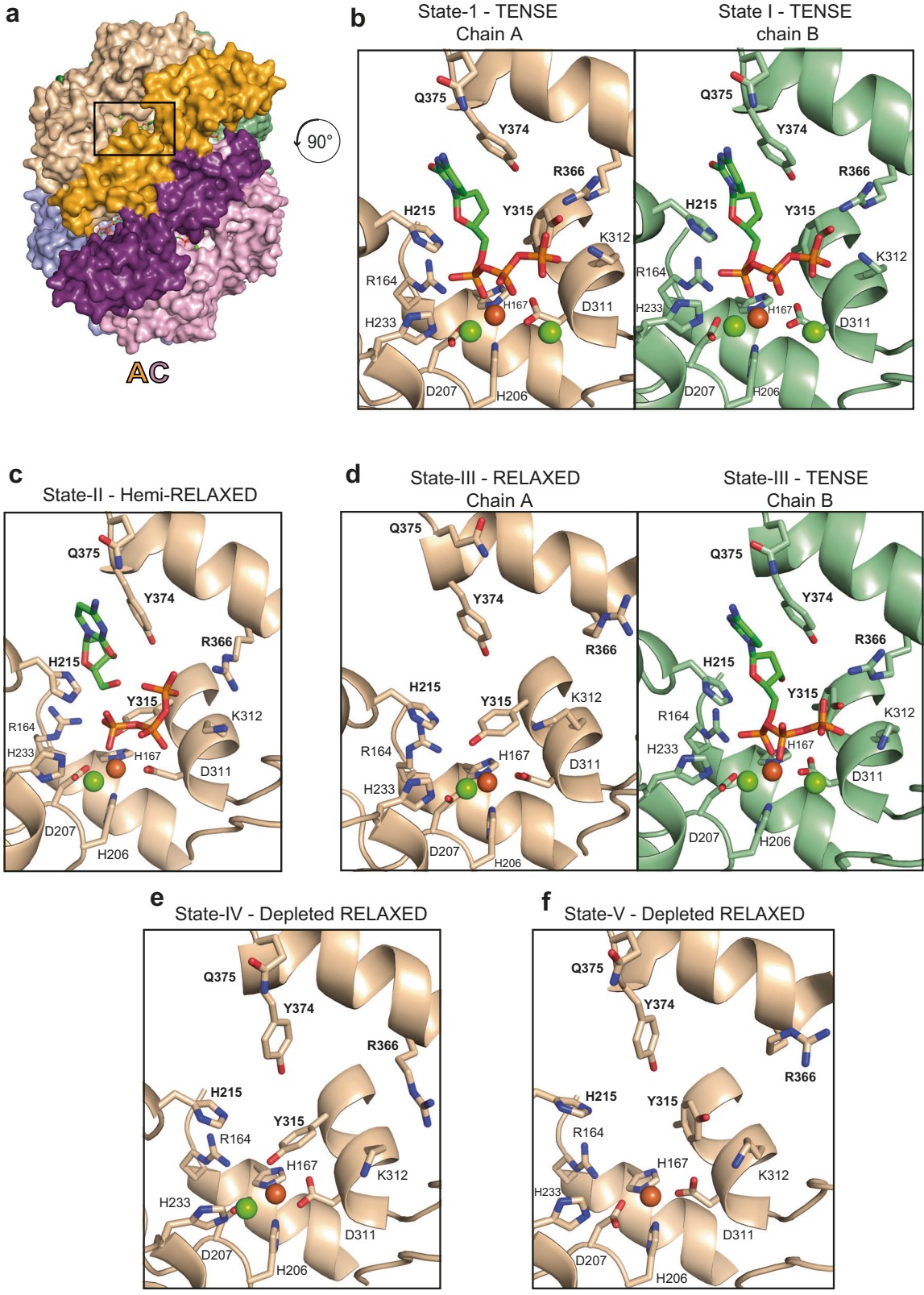

**Fig. 3 | States-I to -V active site occupancy and conformational changes. a** State-I tetramer shown in surface representation, view onto the AC Dimer-2 coloured as in Fig. 1. The nucleotides bound at the active and allosteric sites are shown in stick representation, the active site of Monomer-A is boxed. Active site of Monomer-A in (**b**) State-I, (**c**) State-II, (**d**) State-III, (**e**) State-IV and (**f**) State-V. Views are rotated 90° with respect to the box orientation in (**a**), illustrated by the arrowed circle. In (**b**, **d**), the active site of Monomer-B that retains the tense state is shown for comparison. In all panels the protein backbone is shown in cartoon representation, the bound nucleotide, modelled as dCTP, and selected active site residues are shown in stick representation, metal ions as spheres, coloured by atom type (Mg, green; Fe, brown). Residues that adopt different conformations in response to dNTP binding and product release H215, Y315, R366, Y374 and Q375 are labelled in bold.

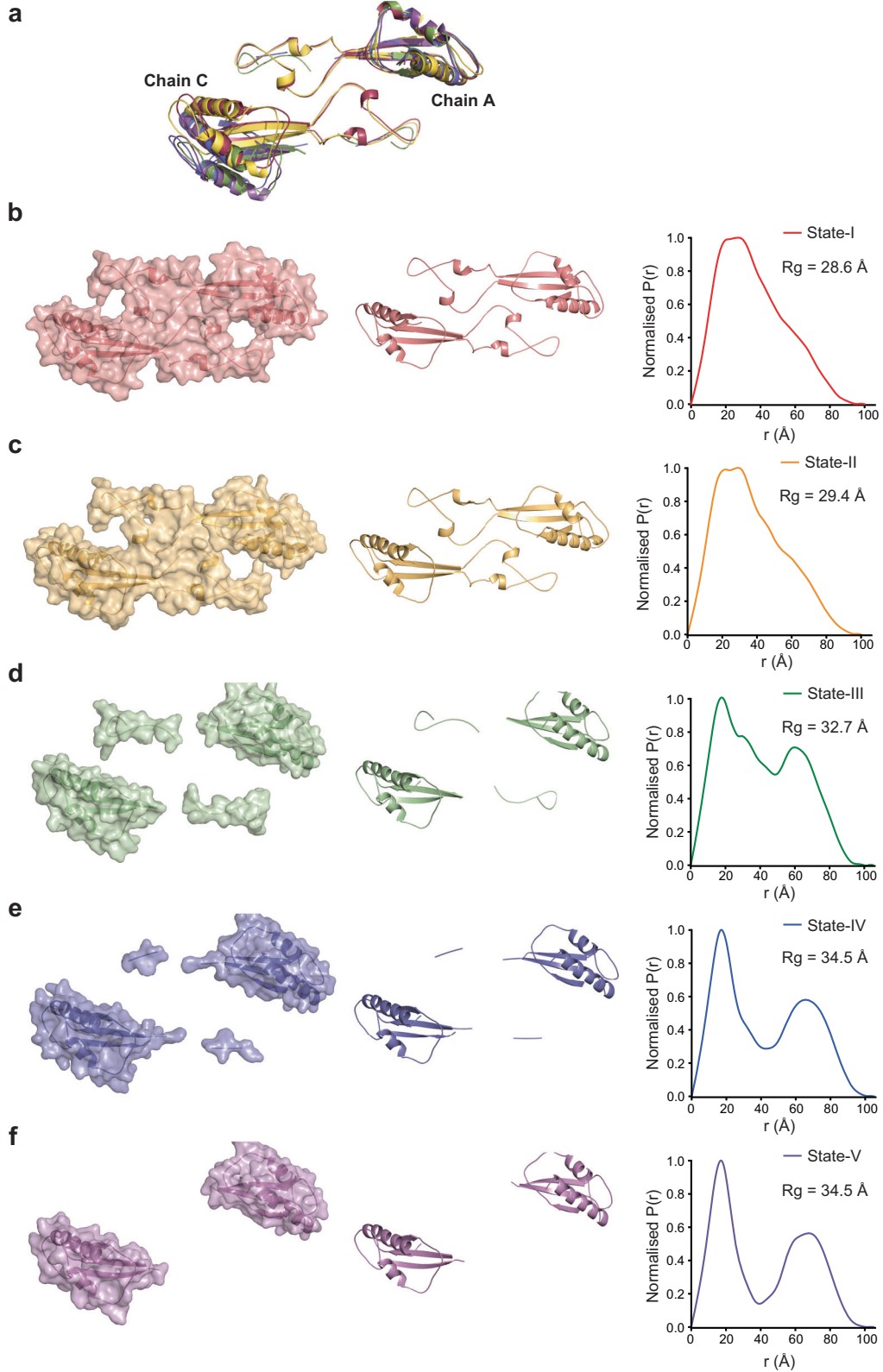

**Fig. 4 | Regulatory domain lobe-linker order-disorder transition.** (**a**) Structural superposition of the Monomer-A and Monomer-C regulatory domain C-terminal lobe and lobe linkers of State-I to -V. The structures, shown in cartoon representation coloured red (State-I), Yellow (State-II), green (State-III), blue (State-IV) and purple (State-V). The structures are aligned on the Monomer-A regulatory domain to show the displacement of the Monomer-C regulatory domain. (**b-f**) C-terminal lobe and lobe linker for the Monomer-A/Monomer-C Dimer-2 of (**b**) State-I, (**c**) State-II, (**d**) State-III, (**e**) State-IV and (**f**) State-V. The left hand and central panels show the ordered region in each state in surface and cartoon representation respectively. The right-hand panels are the corresponding pair distribution function P(r) of interatom vectors derived from Crysol and Gnom analysis of each model. The extent of the function on the x-axis gives the maximum dimension ($D_{max}$) of the particle. The real-space $R_g$ derived from the P(r) function is displayed inset.

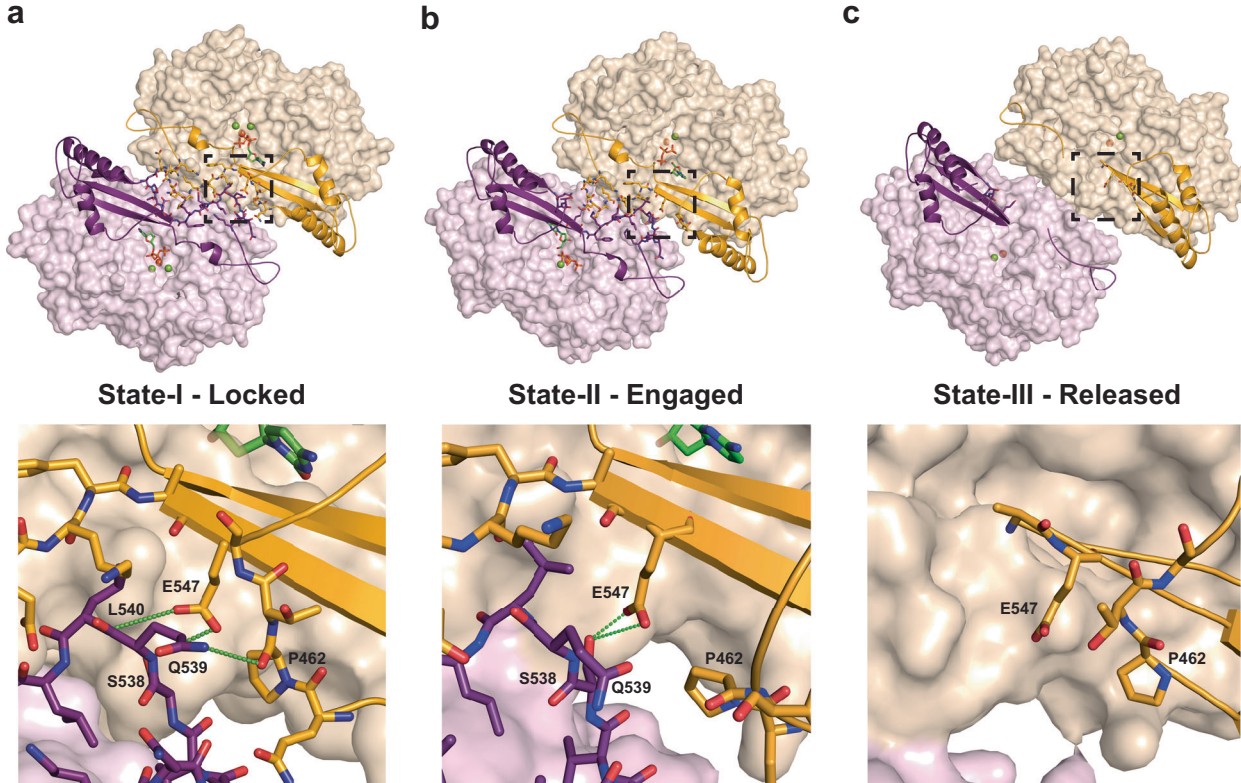

**Fig. 5 | C-terminal lobe coupling to catalysis. a** State-I (**b**) State-II and (**c**) State-III AC Dimer-2 lobe-linker interfaces. In the top panels, catalytic and regulatory domains are shown in surface representation and cartoon representation respectively, coloured as in Fig. 1. Residues at the lobe linker interfaces together nucleotide and nucleosides bound at the active sites are shown in stick representation with metal ions as spheres. The lower panels show the boxed region of the upper panels detailing the lobe-linker - C-terminal lobe interactions in State-I, State-II and State-III that constitute the locked, engaged and released states. Residues at the lobe-linker C-terminal lobe interface are shown in sticks hydrogen bonds are displayed as green dashed lines.

and product release through conformational switches that are transduced into C-terminal lobe motions and order-disorder transitions in the 507-545 region. Measurement of $k_{cat}$ for mutants R164A, H215A, R366A, Q375A and Q375N (Supplementary Fig. 13 and Supplementary Table 6) reveal that all the mutants reduce activity but to substantially different degrees. The transducing residue mutants Q375A and Q375N result in a 15 to 20-fold reduction in $k_{cat}$, R366 diminishes $k_{cat}$, around 300-fold. By contrast, the substrate binding mutant R164A and catalytic residue mutant H215A reduce $k_{cat}$ to below the limit of detection for the assay and in complementary SEC-MALLS analysis of tetramerisation (Supplementary Fig. 14), all mutants with the exception of R164A were assembly competent. These data reveal the importance of all three classes of active site residue that through a combination of substrate binding, catalytic chemistry, and transduction of quaternary conformational effects are all required for efficient catalysis.

**C-terminal lobe order-disorder transition in catalysis**
Within SAMHD1 HD-tetramers each monomer can be divided into catalytic and regulatory domains (Supplementary Fig. 7). The catalytic domain comprises residues 113 to 454 and contains the active site, the AL1 GTP-binding residues and the AL2 dNTP-binding residues excepting K523. The regulatory domain comprises residues 455-599 that includes the C-terminal lobe, residues 455–506 and 546 to 599, as well as the interspersing extended lobe-linker regions, residues 507-545 that form a canopy covering the α13-helices of the Monomer-A/Monomer-C and Monomer-B/Monomer-D State-I Dimer-2 interfaces. It is these extended lobe-linker regions that undergo extensive conformational rearrangement becoming progressively more disordered in the relaxed Dimer-2 pairs from State-I to State-III in the catalytic cycle and further disordered in the nucleotide-depleted State-IV and -V

(Fig. 4, and Supplementary Fig. 15). Associated with these disorder-transitions, the C-terminal lobes of relaxed Monomer-A/Monomer-C Dimer-2 pairs roll away from Dimer-2 two-fold axis by ~10° rotation and increase in separation to a 5–7 Å greater centre-to centre distance between State-I and State-V (Fig. 4a). The combined effect of this disordering of extended lobe-linker regions and C-terminal lobe separation is to both open the catalytic sites and expose the nucleotides at the allosteric sites on the relaxed side of the tetramer (Supplementary Fig. 11).

To assess the degree of disorder and quantify the magnitude and direction of the quaternary conformational changes at each state, we examined the inter- and intra-chain packing within each tetramer. For each chain and isolated regulatory domain, we determined the amount of buried surface resulting from the interaction with the corresponding Dimer-2 mate and also determined the amount of intrachain buried surface for each regulatory domain with its corresponding catalytic domain (Supplementary Table 7). Moreover, for each state of the A-C Dimer-2 we also calculated the radius of gyration ($R_g$) and pair-distribution function, P(r), for all interatomic distances within regulatory domain pairs that reports on the compaction and spacing of the regulatory lobes (Fig. 4b–f). Inspection of the packing of each regulatory domain onto its cognate catalytic domain shows that in the relaxed A-C side it decreases from an average value of 1500 Å² in State I, II and III to 1064 Å² in State-IV and is further reduced to 772 Å in State-V. By contrast, the buried area on the tense side remains at an average value of 1500 Å across all states (Supplementary Fig. 15 and Supplementary Table 7). The calculation for the interchain packing of entire chains and just the regulatory domains against the Dimer-2 mate is even more telling, where in Tense B-D pairs the regulatory domain packing maintains average values across all states from 710 to 602 Å²,

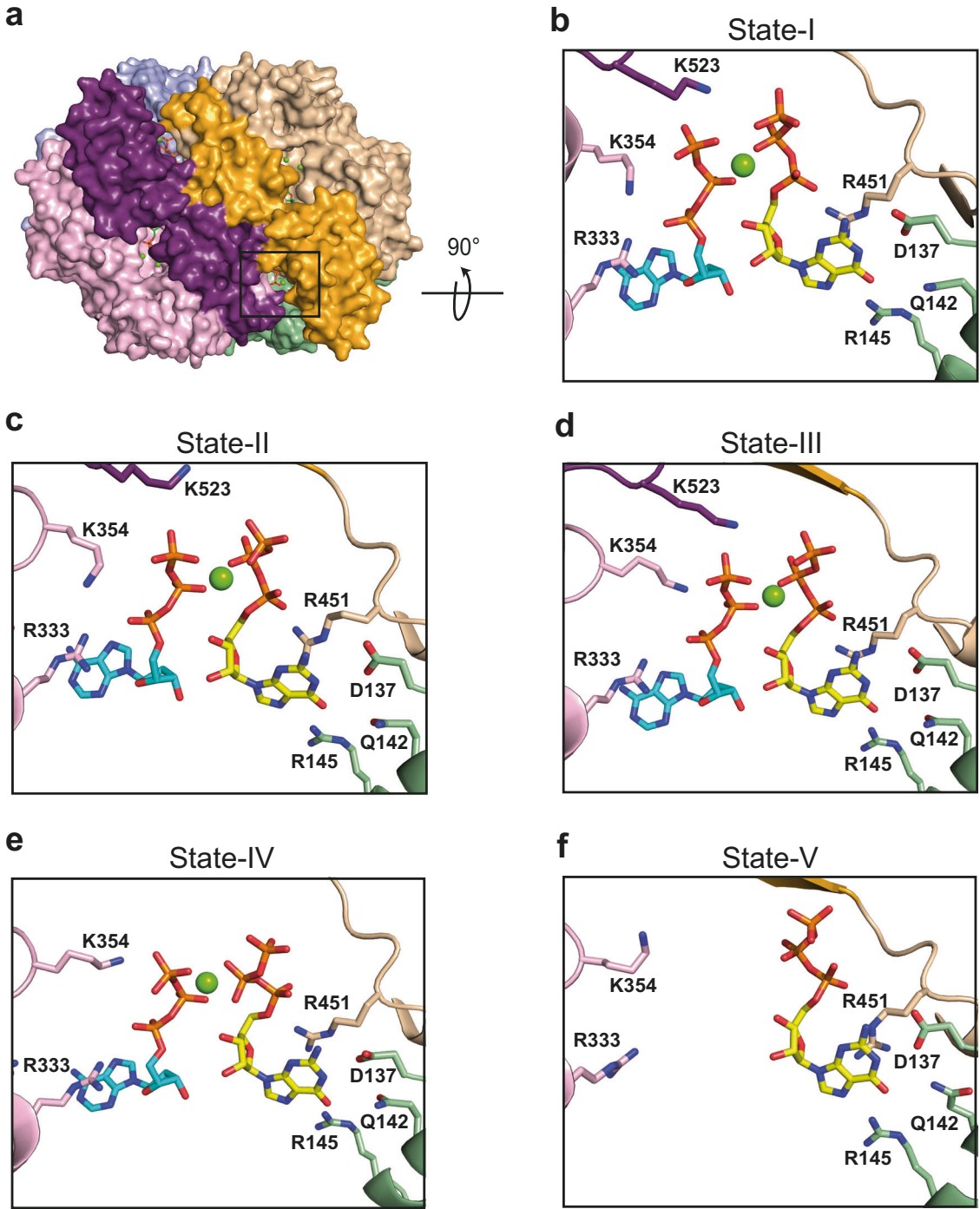

**Fig. 6 | Allosteric site occupancy and conformational changes. a** State-I tetramer shown in surface representation, view onto the AC Dimer-2 coloured as in Fig. 1. The nucleotides bound at the allosteric sites are shown in stick representation, the allosteric site formed at the interface of Monomer-A -B -C (ABC) is boxed. Allosteric site ABC in (**b**) State-I, (**c**) State-II, (**d**) State-III, (**e**) State-IV and (**f**) State-V tetramers.

Views are rotated 90° with respect to the box orientation in (**a**). In each panel the protein backbone is shown in cartoon representation, residues from N-terminal catalytic domains and C-terminal regulatory domains are coloured as in Fig. 1. The bound AL1 GTP (yellow) and AL2 dATP (cyan) nucleotides and selected allosteric site residues are shown in stick representation, Mg ions as green spheres.

whilst in relaxed A-C Dimer-2 pairs this value reduces from 722 and 553 Å$^2$ in State-I and -II to a value of zero resulting from the total uncoupling observed in States III, IV and V (Supplementary Table 7). The same effect is also apparent on inspection of the entire interchain packing where the Tense B-D Dimer-2 maintains an average buried surface area across all states of 1164 Å$^2$ but in the relaxed A-C Dimer-2 it reduces from an average value of 1160 Å$^2$ in State-I and -II to only 350 Å in State-V. These measurements are further supported by the pair-distribution functions, where for States-I to -V there is only a modest

increase in D$_{max}$ of 96 Å to 101 Å relating to an increase in the C-terminal lobe spacing progressing from State-I and State-V (Supplementary Movie 1). However, the functions also display a large very distinct increase in bi-modality that relates directly to the increased degree of disorder of each lobe-linker dimer going from State-I to State-V (Fig. 4). Moreover, this is apparent in the calculated R$_g$ which also reports on compaction and increases from 28.6 Å in State-I to 32.6 Å in State-III and 34.6 Å in State-IV and -V as the linkers become disordered, and the C-terminal lobes separate.

At a molecular level, the coupling of the lobe-linkers to the C-terminal lobes at the Dimer-2 interface is mediated by hydrogen bonding interactions between P462 and E547 on the β7 - α18 interspersing loop and the β11 strand of the C-terminal lobe with S538, Q539 and L540 of the lobe-linker in the opposing monomer (Fig. 5). In State-I this constitutes a locked conformation where the E547 sidechain makes hydrogen bonds to the sidechain hydroxyl of S538 and the backbone amides of both Q539 and L540. In addition, the Q539 sidechain in the lobe-linker reciprocates the interaction through further hydrogen bonding to the backbone carbonyl of P462 in the β7 - α18 loop (Fig. 5a). In state-II the C-terminal lobes move away from the lobe linkers and whilst the E547 sidechain retains the hydrogen bonding to S538, the β7 - α18 interspersing loop is displaced by about 3 Å from its State-I position and the Q539-P462 interaction is no longer present so constituting an engaged conformation (Fig. 5b). In State-III the further movement of the C-terminal lobes away from the lobe-linkers results in loss of all remaining interactions constituting a released conformation (Fig. 5c). In this way the catalytic cycle can be viewed in terms of the transition between locked-engaged-released conformations of the CTD lobes in States-I to III that are intimately coupled with substrate binding, catalysis, and product release.

## Allosteric site configuration and quasi-stable tetramers

Inspection of the nucleotide content of allosteric sites for each state shows that throughout the steady state when States-I, -II and -III predominate, the full complement of GTP/dATP nucleotides and bridging Mg1 are present at the AL1 and AL2 sites in both the relaxed A-C and tense B-D Dimer-2 pairs (Fig. 6a–d, Fig. 2a–c and Supplementary Table 4). This suggests that although there are dynamic domain motions that result in some exposure of the nucleotides at allosteric binding sites, it does not result in nucleotide loss, supporting the notion that there is no requirement to reload nucleotides at the allosteric sites during the steady state. At the later stages of the reaction >2400 s when substrate nucleotides are >99% depleted, the State-IV and -V tetramers predominate. Inspection of these allosteric sites show that state-IV tetramers still have a complement of AL1 and AL2 nucleotides in both the tense and relaxed sides of the tetramer, albeit the density is weaker for the dATP in the relaxed side (Figs. 6e and 2d). It is only in State-V where loss of nucleotides from the AL2 site is apparent in the relaxed side (Figs. 6f and 2e) although the GTP is maintained bound at AL1. These data show that under conditions of nucleotide depletion SAMHD1 tetramers still persist as has been observed in solution experiments[16,55]. Furthermore, our data now show that filling of allosteric sites produces an underlying stable platform of tetrameric catalytic domains that in turn supports the dynamic motions of the associated regulatory domains that control the catalytic cycle and give rise to the differing tetramers observed at the steady state. In conditions of nucleotide depletion, it is also apparent that the stable tetrameric platform of catalytic domains is maintained even with partial nucleotide loss from the active and AL2 sites. These quasi-stable State-IV and State-V tetramers are then poised to keep SAMHD1 potentiated in the absence of nucleotides and without the need to fully reassemble the catalytically competent tetramer in response to a rising dNTP level.

## Discussion

Previous solution and structural studies demonstrated that GTP/dGTP promotes SAMHD1 dimerisation and further binding of dNTPs facilitates assembly into the catalytically active tetramer[2,48,55]. Our cEM structures now shed light on this assembly process, definitively showing that Dimer-1 is the building block of the SAMHD1 tetramer. It is present in our GTP-only samples prior to addition of dNTPs and at the very early timepoint, 10 s after dNTP addition, where the Dimer-1 species constitutes about 50% of the total particles with no Dimer-2 species ever observed. It is also at this early part of assembly process

where we observe a protein-concentration dependent lag phase in the enzymatic rate, suggesting further association of preformed GTP-bound Dimer-1 and filling dNTPs to AL2 sites is the rate limiting step of tetramer assembly. The importance of Dimer-1 as the tetramer building block is also apparent at later timepoints as the Dimer-1 conformation that remains invariant in the State-II and State-III tetramers also persists in the nucleotide depleted quasi-stable State-IV and -V tetramers. By contrast, the Dimer-2 conformation is highly variable across State-I to -V, where the progressive disorder and differential packing of the C-terminal lobe-linkers at Dimer-2 interfaces results in the tense and relaxed side asymmetry that is observed in State-II to State-V tetramers.

Our SAMHD1 cEM structures share many similarities with respect to the allosteric and active sites observed in previous structures[16,35,48,52]. However, with regards to the active site residue configuration and quaternary structural changes we observe, these cEM data now provide an understanding of the mechanics and dynamics of SAMHD1 catalysis. To establish this mechanistic model for SAMHD1 catalysis we start with two premises. First, that dNTP hydrolysis occurs within the confines of the D2 symmetrical State-I tetramer where each Dimer-2 pair is in the tense state, substrates are bound at the Fe-Mg bimetallic centre and the phosphate, sugar and base-binding active site residues are engaged with the dNTP. This configuration is observed in both the crystal and our cEM structures of SAMHD1-inhibitor complexes that have the State-I conformation with the attacking nucleophile hydroxyl positioned on the substrate $P^\alpha$ and likely represent the catalytic state prior to adduction and formation of a transition state. Further, we proffer that for new substrates to gain access to the active site, State-I tetramers need to undergo partial or total relaxation, as is observed in the asymmetric State-II to -V tetramers, in order to eject products and reload fresh substrate. Based on our observations and these notions, several possible mechanisms for hydrolysis and enzyme recycling can be conceived. However, in the steady state we consistently observe only closed State-I D2 tetramers along with the State-II and State-III asymmetric tetramers with one relaxed and one tense Dimer-2 pair. Therefore, the model that most consistently satisfies our data is where hydrolysis within the State-I D2 tetramer occurs at two coupled active sites across a Dimer-2 pair, either concertedly or within rapid succession. The concomitant relaxation of the dimer pair that we observe as State-II and State-III then allows product release and further substrate binding to re-establish the substrate-loaded D2 State-I. As we only observe a static image of the asymmetric tetramer, it is not possible to determine if the hydrolysis mechanism utilises a Flip-Flop mechanism where A-C and B-D Dimer-2 pairs alternate in the hydrolysis of dNTPs, or if hydrolysis is a stochastic event resulting from random firing of any pair of Dimer-2 coupled active sites. Regardless, our observations of this molecular machine in action provide valuable insight into the coupling of active sites and the quaternary conformational changes that drive the SAMHD1 catalytic cycle.

As well as informing on the mechanism and dynamics of the SAMHD1 catalytic cycle, our observations also explain how nucleotides bound at the allosteric sites can promote catalysis at the active sites. Our structures demonstrate that the tetramerisation induced by the binding of nucleotides to the allosteric sites is multi-layered. It is directed both through Dimer-2 α13-α13 interactions that promote formation of the catalytic domains into a stable platform and by more dynamic interactions of the regulatory domains that form a fluid layer of interaction covering the catalytic domains, mediated by the lobe-linker interactions with the C-terminal lobes. At one level, allosteric communication is simply a product of the nucleotide-promoted assembly that directs the formation of a stable tetrameric catalytic platform with active sites that remain competent throughout the catalytic cycle without the necessity to undergo large conformational rearrangements, or to disassemble and re-assemble. However, a further consequence of this nucleotide-directed tetramer assembly is to bring

the dynamic regulatory domains into proximity with their dimer-mates and with the active sites. It is then the dynamic motions resulting in the tense and relaxed Dimer-2 pairs that control the catalytic cycle by directing traffic in and out of the active sites through the global order-disorder transitions of the lobe-linkers, and by the local interactions of active site residues with the transducing residues, R366 and Q375. In this way, once assembled the underlying stability of the catalytic platform ensures SAMHD1 activity is always primed and whilst dynamic movements of the regulatory domains control active site access and catalysis, the conformational re-arrangements do not impact on the core tetramer assembly and the capacity for catalysis.

It is also apparent that even when there is a loss of nucleotides at the active sites and at AL2 in State-V, the tetramer still persists through the underlying quasi-stable conformation of tetrameric catalytic domains. Therefore, our data also provide a molecular explanation for the persistence of SAMHD1 tetramers in nucleotide-depleted conditions that enables the enzyme to remain potentiated in the absence of nucleotides[55,16] and has been proposed to be required for SAMHD1 anti-viral function in maintaining low dNTP levels in differentiated myeloid cells.

Many multi-subunit enzymes rely on allosteric communication from regulatory ligand-binding sites to direct catalysis at active sites through subtle changes in protein conformation, stability, and side-chain interaction networks[64–66]. Our visualisation of the SAMHD1 catalytic cycle now advances a concept of dynamic allostery in homo- and hetero-oligomeric systems where binding of allosteric effectors directs subunit assembly resulting in the formation of a stable and catalytically competent core juxtaposed with mobile regulatory domains. Catalysis is then controlled both by large-scale quaternary re-arrangements of regulatory domains as well as short range interactions with peripheral active-site transducing residues that control substrate access, product release and catalysis without disruption to the catalytic core. Given the prevalence of homo- and hetero-oligomeric enzymes containing catalytic and regulatory domains, we suggest this concept of platform-directed dynamic allostery is likely a general mechanism for the regulation of catalysis. The application of further time-resolved structural studies to investigate other homo- and hetero-oligomeric systems will reveal the full extent of this.

## Methods

### Protein expression and purification
Human SAMHD1 residues 1−626 was expressed in *E. coli* from a pET52b vector (Novagen) as an N-terminal StrepII-tag fusion protein[51]. The point mutants R164A, H215A, R366A, Q375A and Q375N were prepared from the parent construct using the Quikchange II kit. Mutagenesis primers were from Merck Life Science, sequences are provided in Supplementary Table 8 and all mutations were verified by DNA sequencing (Full circle Labs). StrepII-tagged SAMHD1 was expressed in the *E. coli* strain Rosetta 2 (DE3) (Novagen) grown at 37 °C with shaking. Protein expression was induced by addition of 0.1 mM IPTG (Melford Biolaboratories) to log phase cultures (A$_{600}$ = 0.5) and the cells incubated for a further 20 h at 18 °C. Cells were harvested by centrifugation, resuspended in 50 mL of lysis buffer; 50 mM Tris-HCl pH 7.8, 500 mM NaCl, 4 mM MgCl$_2$, 0.5 mM TCEP, 1x EDTA-free mini complete protease inhibitors (Roche) per 10 g of cell pellet and lysed by sonication. The lysate was cleared by centrifugation for 1 h at 50,000 x *g* and 4 °C, and subsequently applied to a 10 mL StrepTactin affinity column (IBA), followed by 300 mL of wash buffer (50 mM Tris-HCl pH 7.8, 500 mM NaCl, 4 mM MgCl$_2$, 0.5 mM TCEP) at 4 °C. Bound proteins were eluted from the column by circulation of 0.5 mg of PreScission Protease (Cytiva) in 25 mL of wash buffer over the column in a closed circuit overnight. The supernatant (25 mL) and an additional 15 mL elution of wash buffer were pooled and concentrated to 2.5 mL. Pre-Scission Protease was removed by affinity chromatography using a 1 mL GSTrap HP column (Cytiva) and the eluent was applied to a Superdex 200 26/60 (Cytiva) size exclusion column equilibrated with a gel filtration buffer of 10 mM Tris-HCl pH 7.8, 150 mM NaCl, 4 mM MgCl$_2$, 0.5 mM TCEP. Peak fractions, identified by SDS gel electrophoresis, were concentrated to approximately 20 mg.mL$^{-1}$ and flash-frozen in liquid nitrogen in small aliquots at −80 °C. SAMHD1 active site mutant proteins were purified in the same way and the introduced mutations were verified by ESI-MS.

### Nucleotides
GTP and dNTPs were purchased from ThermoFisher Scientific. The dApNHpp nucleotide analogue was purchased from Jena Bioscience, DE.

### NMR analysis of SAMHD1 catalysis
1D $^1$H NMR spectroscopy was used to measure SAMHD1 dNTP hydrolysis rates. Reactions were prepared in NMR assay buffer (20 mM Tris-HCl pH 8.0, 150 mM NaCl, 5 mM MgCl$_2$, 2 mM TCEP, 5% D$_2$O) containing 0.5 mM of each dNTP, 0.2 mM GTP and 2 μM of SAMHD1. For determination of hydrolysis rates for SAMHD1 mutants 0.2 mM GTP and 0.5 mM dATP with 1–10 μM wt or SAMHD1 mutant was employed. $^1$H NMR spectra (2 dummy scans, 4 scans) pulse sequence PE-WATERGATE[67], were recorded at 30 s intervals as a pseudo 2D array at 293 K using either a Bruker Avance III 600 MHz or Avance IIIHD 700 MHz NMR spectrometer equipped with a 5 mm TCI cryoprobe. Solvent suppression was achieved using excitation sculpting[68]. Experiments were typically carried out for up to 2 h. Spectra were recorded and processed using Topspin 3.6.4 (Bruker). The integrals for clearly resolved substrate and product peaks at each time-point were then extracted using the Bruker Dynamics Center 2.8 software package and used to construct plots of substrate or product against time. Initial rates were extracted from the linear part of the curve in order to determine apparent $k_{cat}$ values.

### Stopped-flow measurements
To measure real-time kinetics of SAMHD1 substrate hydrolysis the coupled assay utilising the biosensor MDCC-PBP[69,70] to measure phosphate (P$_i$) release from combined SAMHD1 triphosphohydrolase and *S. cerevisiae* Ppx1 exopolyphosphatase activity, was employed[71]. In order to resolve the initial lag phase, experiments were performed in a stopped-flow setup using a HiTech SF61 DX2 instrument, equipped with a mercury−xenon lamp (TgK Scientific Ltd). Fluorescence data were recorded with a monochromator on the excitation light (436 nm, 3 mm slit width), and a 475 nm long pass filter (Schott) on the emission. In a typical experiment, SAMHD1, at varying concentrations, was pre-incubated with 0.4 mM GTP for 300 s at 25 °C, before rapidly mixing with an equal volume of 1 mM TTP, both solutions containing Ppx1 and MDCC-PBP. The final concentrations (after mixing) were 0, 0.5, 1, 2, 4 or 8 μM SAMHD1, 0.2 mM GTP, 0.5 mM TTP, 0.1 μM Ppx1 and 40 μM MDCC-PBP. Experiments were performed in assay buffer (20 mM Tris pH 8.0, 150 mM NaCl, 5 mM MgCl$_2$ and 2 mM TCEP) at 25 °C. For calibration, MDCC-PBP was mixed with a series of P$_i$ concentrations (0 to 15 μM final) in the stopped-flow and the recorded fluorescence intensity was plotted against P$_i$ concentration. The slope from the linear calibration curve was used to convert the measured fluorescence intensities into μM P$_i$. Maximal rates and lag phases were obtained by linear regression of the time courses of P$_i$ formation in the linear phase of the reaction using the software package Graphpad Prism 9. All measurements were performed in at least triplicate.

### SEC-MALLS
Size Exclusion Chromatography coupled to Multi-Angle Laser Light Scattering (SEC-MALLS) was used to determine the capacity for tetramerisation of SAMHD1 samples upon addition of dApNHpp nucleotide analogue and GTP. 30 μM wt-SAMHD1(1−626) or mutants were incubated with 0.5 mM dApNHpp and 0.2 mM GTP at room temperature for 300 s. Samples (100 μL) were then applied to a

Superdex 200 10/300 INCREASE GL column equilibrated in 20 mM Tris-HCl pH 7.8, 150 mM NaCl, 5 mM $MgCl_2$, 0.5 mM TCEP and 3 mM $NaN_3$ at a flow rate of 1.0 mL.min⁻¹. The scattered light intensity and protein concentration of the column eluate were recorded using a DAWN-HELEOS-II laser photometer and an OPTILAB-TrEX differential refractometer (dRI) ($dn/dc = 0.186$) respectively. The weight-averaged molecular mass of material contained in chromatographic peaks was determined using the combined data from both detectors in the ASTRA software version 7.3.2 (Wyatt Technology Corp., Santa Barbara, CA).

### Cryo-EM sample preparation and data collection

SAMHD1 (20 μM) was premixed with 0.2 mM GTP in assay buffer at 25 °C prior to the addition of deoxynucleotides by dilution of the sample into a final reaction mix containing 2 μM SAMHD1, 0.5 mM of dATP, dCTP. dGTP and TTP and 0.2 mM GTP or 0.5 mM dApNHpp only to form the inhibitor complex. Samples were then further incubated at 25 °C for between 10 and 5400 s before application to grids, either UltrAuFoil® R2/2 200 or Quantifoil Cu R2/2 200 mesh prepared by glow discharge at 25 mA for 60 s or 30 s respectively in air (EMITECH). An additional T = 0 s sample that contained only SAMHD1 and 0.2 mM GTP was frozen in the same way but without adding further deoxynucleotides. All grids were frozen using a FEI Vitrobot mark III at 4 °C and 100% relative humidity. 3 μL sample was added to carbon side of the grid and immediately blotted for between 3.5 and 5.0 s before plunge freezing into liquid ethane. Data from frozen-hydrated samples were collected on a Titan Krios operating at 300 kV (ThermoFisher Scientific) in nanoprobe mode. Datasets were either collected using a Gatan GIF Quantum energy filter in zero-loss mode, slit width of 20 eV and Gatan K2 detector operating in counting mode with a calibrated pixel size of 1.08 Å or on a Falcon III detector operating in counting mode with a calibrated pixel size of 1.09 Å. Movies were recorded with an exposure time of 9.0 s, corresponding to 30 frames and a total dose of 48.6 e⁻/Å² (Gatan K2) or 60 s corresponding to 30 frames and a total dose of 33.0 e⁻/Å² (Falcon III). All datasets were collected with defocus range of −1 to 3.5 μm.

### Image processing

All movies were motion-corrected, CTF parameters estimated, particles picked and extracted (Binned by 2) on-the-fly using Warp 1.0.9[72]. Micrographs with a CTF quality of fit better than 7 Å resolution were selected for further analysis. Particles were imported into cryoSPARC v2[73,74] and subjected to two rounds of 2D classification to identify particles of high quality and an initial 3D model was derived using the ab initio reconstruction tool. Particles and the initial model were then imported into Relion 3.1[75,76] where 2 rounds of 3D classification with 25 iterations each were undertaken. In the first round with angular sampling intervals of 7.5° and in the second with 1.8° intervals incorporating local searches. Particles corresponding to fully assembled tetramers were then re-extracted un-binned, imported into CryoSPARC v2 and refined using non-uniform refinement to generate a consensus map.

For the identification of States I–V, the whole stack from each time point was analysed using the 3DVA routine implemented in CryoSPARC v2 using a soft mask around the entire tetramer. Particles were extracted in Cluster mode to generate 10 clusters. Clusters belonging to the same class were pooled and stacks of particles from each class were then refined using non-uniform refinement CryoSPARC v2 first without imposing any symmetry. Finally, particles and the maps were then subject to a final round of non-uniform refinement, also optimising for per particle defocus and CTF parameters with either the application of D2 (Inhibitor & State-I), C2 (States II−IV) and C1 or C2 symmetry for State-V. Reported resolutions are based the gold standard Fourier shell correlation (FSC) with the cut off criterion of 0.143[77]. FSC curves were corrected for the artefacts of soft-masking[78].

### Model building

SAMHD1 monomer structures were initially docked into the inhibitor and State-I to State-V maps using rigid body refinement in Chimera 1.1.6[79]. The fits were then further optimised using the JiggleFit tools and any missing sections of the model built into density in COOT 0.9.8.8[80] with iterative rounds of real space refinement in PHENIX 1.21.1[81]. Whole particle structures were then further refined using PHENIX real-space-refine to produce final refined models for each map. All models comprise residues 113−576 with varying completeness in the regions N577-P589 and D506 - E547 depending on the degree of disorder in the C-terminal lobe-linkers and with no density for the SAM domain (residues 1- 109) or C-terminal Vpx binding region (residues 590−626). The local resolution of maps was determined using Relion 3.1 or ResMap[82]. Models were validated throughout refinement using MolProbity[83], quality of fit was assessed using map vs model FSC in PHENIX and calculation of backbone, sidechain and nucleotide Q-scores[84] implemented in ChimeraX 1.7.1[85]. Additionally, validation of the State-I to State-V models was undertaken in PHENIX by determining the cross-correlation of every chain and nucleotide in each model against the State-I to -V maps. Analysis of buried surface at protein interfaces was undertaken using PISA[86]. Pair distance distribution functions and radius of gyration ($R_g$) for regulatory domain dimers were computed in GNOM[87] from the back-transformed scatter curve of each built atomic model, calculated using CRYSOL[88] within the ATSAS 3.2.1 programme suite[89]. Details of data collection and model refinement are presented in Table 1 and the cross-correlation validation in Supplementary Table 3.

### Reporting summary

Further information on research design is available in the Nature Portfolio Reporting Summary linked to this article.

## Data availability

All atomic models and density maps employed by or generated in this study have been deposited in the Protein Data Bank and Electron Microscopy Data Bank repositories with accession codes: 8QXJ (SAMHD1-inhibitor), EMD-18729 (SAMHD1-inhibitor); 8QXK (SAMHD1 State-I), EMD-18730 (SAMHD1 State-I); 8QXL (SAMHD1 State-II), EMD-18731 (SAMHD1 State-II); 8QXM (SAMHD1 State-III), EMD-18732 (SAMHD1 State-III); 8QXN (SAMHD1 State-IV), EMD-18733 (SAMHD1 State-IV); 8QXO (SAMHD1 State-V), EMD-18734 (SAMHD1 State-V); 6XT0 (SAMHD1-dAMPNPP). The enzymological data in Supplementary Fig. 1, 6 and 13 that support the studies' findings are provided in the Source Data file. Correspondence and requests for materials should be addressed to P.B.R. (peter.rosenthal@crick.ac.uk) and I.A.T. (ian.taylor@crick.ac.uk) Source data are provided with this paper.

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

## Acknowledgements

We gratefully acknowledge the Francis Crick Structural Biology Science Technology platform. We thank members of the Rosenthal and Taylor Laboratory for advice and support and Jonathan Stoye for critical reading of the manuscript. The work was supported by the Francis Crick Institute, which receives its core funding from Cancer Research UK (CC2029, I.A.T.; CC2106, P.B.R.) the UK Medical Research Council (CC2029, I.A.T.; CC2106, P.B.R.) and the Wellcome Trust (CC2029, I.A.T.; CC2106, PBR). NMR spectra were recorded at the MRC UK Biomedical NMR Facility, Francis Crick Institute which is funded by Cancer Research UK (CC1078, G.K.), the UK Medical Research Council (CC1078, G.K.), and the Wellcome Trust (CC1078, G.K.). For the purpose of Open Access, the author has applied a CC BY public copyright licence to any Author Accepted Manuscript version arising from this submission.

## Author contributions

All authors contributed to the design and data analysis of experiments. O.J.A. and A.N. performed cEM studies. O.J.A., D.S. and I.A.T. interpretated maps and built structural models. O.J.A., S.K., S.J.C., A.J.O.B., E.R.M. and G.K. performed biochemical, enzymological and NMR studies. P.B.R. and I.A.T. wrote the manuscript with contributions from O.J.A., D.S., E.R.M. and S.K.

## Funding

## Competing interests

The authors declare no competing interests.
