## [Peer Review File · Nature Communications]

Platform-directed allostery and quaternary structure dynamics of SAMHD1 catalysisREVIEWER COMMENTS

Reviewer #1 (Remarks to the Author):

In this manuscript Acton et al., present comprehensive findings on the assembly process of SAMHD1 tetramers and their role in catalytic activity. The manuscript provides a robust framework employing cryo-electron microscopy (cEM) and time-resolved reactions to investigate the dynamic nature of SAMHD1 tetramer assembly and catalysis. The proposed model for SAMHD1 catalysis, rooted in structural observations and time-resolved analyses, highlights the pivotal role of the State-I D2 tetramer in dNTP hydrolysis, product release, and substrate binding. Furthermore, the study unveils the allosteric-driven tetrameric assembly and explains how dynamic regulatory domains control active site access, elucidating SAMHD1's persistence even in nucleotide-depleted conditions. This integrated cryoEM and time-resolved reaction workflow not only advances understanding of SAMHD1 but also sets a robust framework applicable to exploring dynamic mechanisms in other complex enzyme systems. The data processing approach is on par with the most advanced data processing workflows of the field, and the volume of solved states and their detail is really impressive. The quality of data presentation is good, but can be improved by the suggestions listed below in this review. The conclusions are appropriate and based on the structural findings produced by Acton and colleagues. Text is clearly written, and contextual information based on previous literature is well introduced. I recommend this manuscript to be accepted with minor revisions summarized below.

Line 74: Specify what "order of preference" means in this context (Kd? Ka? Other?)

Line 115: Add time in minutes (besides the seconds) for 300, 600, 2400s and 5400s so that reader doesn't have to do mental (or calculator) math to contextualize results. Eg: 300 s (5 minutes), 600 s (10 minutes)...

Figure 2: The black square in (a) is not oriented in the same way as the insets in b-f. Please add visual representation of how the square is rotated from a to b-f to put the results into the right spatial context and help comprehension by the reader.

Figure 4: a-c panels are not informative at this scale (details are too small to be seen and/or interpreted), moreover a-c it is superfluous to figure 3.

Figure 5: Same issue as Figure 2.

Supplementary figure 4: Consider moving this figure to the main figure session. It is a key figure to understand and contextualize the results and shouldn't be buried in the supplemental session.

Supplemental Figure 12 is more referenced in the text than main Fig. 3. Maybe both figures can be combined or switched.

Supplemental Movie 1: By far the most informative manner to interpret and summarize the findings. Please add labels to illustrate the different states as they cycle through the movie.

Reviewer #2 (Remarks to the Author):

Platform-directed allostery and quaternary structure dynamics of SAMHD1 catalysis by Acton et al. The authors performed cryoEM data coupled to NMR and functional studies to follow dNTP hydrolysis by SAMHD1 in real time. The study is important and the methodology followed by the authors is interesting. However, several technical issues including the quality of the EM data and the overall presentation of figures have a negative impact.

I) EM data:

- 1) The number of particles used for the different stages varies significantly, the resolutions of the data sets of different stages varies from 2.65 Å to 3.95 Å. It is understandable that the later stages have less well-defined regions, however an attempt to improve the resolution could definitely validate the results.
- 2) The maps used to depict the density for the substrates and metals uses an extremely low sigma level (1.5) for an EM map which would make hard to differentiate from the noise level of the map. Are the maps somehow normalized (cut) to a small box?
- 3) It is not clear how the authors could differentiate the Fe vs Mg metals, only the maps for the allosteric sites show clear density for the Mg ion.
- 4) cryogenic-EM (cEM)", cryo-EM is the standard term.
- 5) Time-point T=10s, is actually taken after 30 sec of incubation on the grid (4 C) and some extra time for solution application, so is it actually ~60s after reaction initiation.
- 6) According to Methods section, state V was refined to C1 or C2 symmetry. Was there any difference? Did the resolution of all areas improve by imposing D2 and C2 symmetry in other structures?
- 7) It is often the case in multimeric proteins that not all active sites are similar since the substrate occupancy might vary in each pocket. The authors should analyze how pockets vary for each state
- 8) Overall, the authors need to validate their result with better quality maps and improve the resolution of the data.

II) Figures and text.

Figure 1 is difficult to understand. Depicting the tetramers as surface representations might not highlight the differences between the states; for example, comparisons between panel a (tense) and hemi-relaxed, panel c, are completely not obvious. The authors need to show perhaps a cartoon representation overlaying the different states and zooming in on the substrates, and the allosteric sites, which are practically invisible in the figure. Are there any differences in the overall dimensions (length and width) of the tetramer between the tense and relaxed structures and between states. The figure legend is huge, needs to be summarized better.

Line 123, it is hard to quantify the metal content at this resolution.

Supplementary figure 2 doesn't show the 3-D class for the dimer.

Supplementary figure 3 (panels a and b) should show the resolution for each state.

Supplementary figure 6 shows the topology of a monomer observed in previous crystal structures, what is the point. Authors should show the topology of the different cryoEM states and compare with the

crystal structures. Perhaps this depiction could allow the reader to understand where the monomer/dimer interactions take place.

Lines 149 to 155, where are all these residues and interactions illustrated.

Supplementary figure 7, colors are hard to differentiate.

Line 160. authors should show how dimer 2 does not fit in the density.

Supplementary figure 8 shows the best density maps in the presence of the non-hydrolyzable analog; however, density for metals in panel e is not clear.

Line 172, what does “modeled” mean?

Lines 175-179, authors show illustrate this arrangement with a cartoon.

Lines 180-184, not clear where these residues and interactions are located, should be shown.

Lines 183-185, authors should overlay cartoon representations to give the reader a sense of the differences between the tense, relaxed and semi-relaxed structures.

Supplementary figure 9, is one of the most relevant figures of the paper, since it validates the results of Figure 2. Panel a shows density at the active site for the substrate and the metals. This should be made clear with a better depiction of the maps. As mentioned before, 1.5 sigma is way too low. It ought to be shown that this density is not noise. Can they base type be identified? The active site of panel b does not illustrate the product. The density is not clear at all. It is hard to see how the metals in panels c-e would stay in place; maps illustrating the coordinating residues and distances should be included to validate the results.

Figure 2 needs validation with maps. H-bond distances and coordination distances of metals should be indicated to validate lines 213-216.

Lines 216-225 need map validation; the resolution is too low to make these conclusions.

Line 226, the authors should point to supplementary figure 10H (also line 235), which depicts these conformational changes.

Lines 228-231 is this true for each monomer?

Figure 2 should include overlay of residues to illustrate conformational changes mentioned in lines 237-239.

Figure 3 should show how the catalytic sites are exposed during the order/disorder transition in catalysis (line 277).

Figure 4. There is no good color contrast for panels a-c. Panels d-f should show the substrate and how the locked-engage-release conformers relate to substrate binding, catalysis and release.

Discussion should include a schematic of the proposed cycle, from assembly to substrate binding, catalysis and release.

Reviewer #3 (Remarks to the Author):

Acton et al, Platform-directed allostery and quaternary structure dynamics of SAMHD1 catalysis.

SAMHD1 is deoxynucleotide triphosphate hydrolase and a critical regulator of cellular dNTP homeostasis. The authors have used time-resolved cryoEM to visualize the assembly and allostery of

SAMHD1. They were able to observe how conformational changes in the tetramer drive the catalytic cycle. Specifically, they identified 5 different conformational states and propose that opening and closing of the active sites drives enzyme catalysis.

This is an excellent manuscript thoroughly describing the detailed mechanism of SAMHD1 that will be of relevance to those interested in nucleotide metabolism, cell cycle regulation or multi-subunit enzyme regulation. The figures are clear and well-presented and the conclusions drawn from the data are logical. The structures seem to be well determined and the additional data solid.

I only have a couple of very minor comments for the authors' consideration:

In the first sentence of the introduction, the authors state that SAMHD1 is a dNTPase and provide a reference. I think it would be more appropriate to have the original description of the dNTPase activity added to the references (including the authors own paper) PMID: 22056990 and PMID: 22069334

Pg 5 Line 78, The authors state that phosphorylation regulates dNTPase activity. This idea is fairly controversial, and there is significant data that suggests otherwise. The authors may want to modify or qualify the statement.

It is curious that the SAM domain does not have any density in any of the structures. Just curious if the authors have thoughts about if or how the SAM domain may contribute to the structural changes or conformational states observed.

Response to reviewers: manuscript NCOMMS-23-53244A Acton et al 2023

Reviewer-1

In this manuscript Acton et al., present comprehensive findings on the assembly process of SAMHD1 tetramers and their role in catalytic activity. The manuscript provides a robust framework employing cryo-electron microscopy (cEM) and time-resolved reactions to investigate the dynamic nature of SAMHD1 tetramer assembly and catalysis. The proposed model for SAMHD1 catalysis, rooted in structural observations and time-resolved analyses, highlights the pivotal role of the State-I D2 tetramer in dNTP hydrolysis, product release, and substrate binding. Furthermore, the study unveils the allosteric-driven tetrameric assembly and explains how dynamic regulatory domains control active site access, elucidating SAMHD1's persistence even in nucleotide-depleted conditions. This integrated cryoEM and time-resolved reaction workflow not only advances understanding of SAMHD1 but also sets a robust framework applicable to exploring dynamic mechanisms in other complex enzyme systems. The data processing approach is on par with the most advanced data processing workflows of the field, and the volume of solved states and their detail is really impressive. The quality of data presentation is good, but can be improved by the suggestions listed below in this review. The conclusions are appropriate and based on the structural findings produced by Acton and colleagues. Text is clearly written, and contextual information based on previous literature is well introduced. I recommend this manuscript to be accepted with minor revisions summarized below.

We thank Reviewer-1 for their appreciation of the manuscript.

Line 74: Specify what "order of preference" means in this context (K_d? K_a? Other?)

Order of preference is based on affinity with respect to occupancy in crystal structures and apparent K_A from AL2 affinity measurements. I have inserted K_A in the sentence to clarify.

Line 115: Add time in minutes (besides the seconds) for 300, 600, 2400s and 5400s so that reader doesn't have to do mental (or calculator) math to contextualize results. Eg: 300 s (5 minutes), 600 s (10 minutes)...

I have now included the time in minutes in the text.

Figure 2: The black square in (a) is not oriented in the same way as the insets in b-f. Please add visual representation of how the square is rotated from a to b-f to put the results into the right spatial context and help comprehension by the reader.

I have amended the legend of Figure 2 (now Figure 3) to state that the views in panels **b-f** are rotated 90° with respect to the box orientation in panel **a**. Additionally, I have now included an orientational arrow on the figure.

Figure 4: a-c panels are not informative at this scale (details are too small to be seen and/or interpreted), moreover a-c it is superfluous to figure 3.

We appreciate the original a, b and c panels are small. However, we feel their main purpose is to orient the reader in the structure with respect to the lower panels that contain the H-bonding details of the C-terminal lobe – linker interactions. We have maintained the figure (now called Figure 5) but now edited it to combine panels so that panel **a** includes old panel **d**, **b** includes old panel **e**, and **c** includes old panel **f** and also has boxes around the regions of interest in the top panels that relate directly to the bottom panels. The legend has also been amended to connect the relationship more fully between each upper and lower panel. We have also replaced the old Figure 3 with the much more detailed Supplementary Figure 12.

Figure 5: Same issue as Figure 2.

I have amended the legend of Figure 5 (now Figure 6) to state that the views in panels **b-f** are rotated 90 ° with respect to the box orientation in panel **a**. I have also added an orientational arrow to the figure.

Supplementary figure 4: Consider moving this figure to the main figure session. It is a key figure to understand and contextualize the results and shouldn't be buried in the supplemental session.

We agree on the importance of the state data and so have now moved Supplementary Figure 4 to panel **a** of an updated Figure 1.

Supplemental Figure 12 is more referenced in the text than main Fig. 3. Maybe both figures can be combined or switched.

We have now switched Supplementary Figure 12 into the main text to become a new Figure 4.

Supplemental Movie 1: By far the most informative manner to interpret and summarize the findings. Please add labels to illustrate the different states as they cycle through the movie.

We have now included text labels to accompany each state of SAMHD1 as the states are cycled through the movie. In addition, given its illustrative contribution to the manuscript, we have now referenced the movie much earlier in the text when we first introduce the five states.

Reviewer-2

Platform-directed allostery and quaternary structure dynamics of SAMHD1 catalysis by Acton et al. The authors performed cryoEM data coupled to NMR and functional studies to follow dNTP hydrolysis by SAMHD1 in real time. The study is important and the methodology followed by the authors is interesting. However, several technical issues including the quality of the EM data and the overall presentation of figures have a negative impact.

We thank Reviewer-2 for recognising the importance of the study and suggesting improvements in presentation as well as questions on the EM data. The EM data unequivocally support our conclusions with the appropriate information content in the maps and we have used established validation criteria. We believe the answers to the questions below will help clarify this. We think the referee's concerns are related to how we have presented the data in figures rather than a problem in our data. We appreciate that the figures in the paper focus more on the results rather than demonstrating the agreement between our models and maps and have made figure changes reflecting comments by this and the other referees.

1) EM data:

1) The number of particles used for the different stages varies significantly, the resolutions of the data sets of different stages varies from 2.65 Å to 3.95 Å. It is understandable that the later stages have less well-defined regions, however an attempt to improve the resolution could definitely validate the results.

A virtue of our approach is that we are visualising a live reaction so we cannot expect our states to have the same number of particles nor the same resolution due to different

amounts of disorder as the referee points out. We have made claims about each map and feature as is appropriate for the resolution.

In response to this point, we have carefully reviewed all the claims in the manuscript. The structural features in our maps substantiate our interpretation. We have validated the maps using all standard criteria and also included additional specific assessments which we felt were important in the context of this specific study (Table 1, Supplementary Table 2, and Figure 2 & Supplementary Figure 8). We have also now carried some reprocessing of the data, with respect to symmetry checking and improving the resolution see response to points 6, 7 and 8. Some of the other concerns the referee had may be related to point 2, where we reported low sigma values for small boxes as well as the identity of the metal ions in point 3, see responses below.

The main limitations of our findings have already been discussed in the conclusions and we have been precise about this: regarding the static picture we see of the C2 tetramers, indeed there could be additional states in addition to or intermediate between those observed. Nevertheless, our structural understanding of SAMHD1 catalysis represents a significant advance as recognised by the referees and our understanding within our current approach is largely complete.

2) The maps used to depict the density for the substrates and metals uses an extremely low sigma level (1.5) for an EM map which would make hard to differentiate from the noise level of the map. Are the maps somehow normalized (cut) to a small box?

We thank the referee for this observation. Because of the known dependence of sigma on box size as the referee indicated, we did not see this in itself as a problem. However, we have recalculated sigma values for the full deposited map box, and this is now more in line with the expectation of users of the maps. E.g. our ligands have sigma values $\sim 8 - 15$. We believe this may address the concern by the referee regarding our data quality as a whole. The maps themselves are available to the referee as part of the review process.

3) It is not clear how the authors could differentiate the Fe vs Mg metals, only the maps for the allosteric sites show clear density for the Mg ion.

In our highest resolution maps (State-I and inhibitor), we see distinct separate densities corresponding to ions that are in the same positions as ions in high-resolution crystal structures that we have determined previously, 6TX0, 6XU1 and many others (see Morris et al 2020 Nat Commun 11, 3165 & Arnold et al 2015 Plos Pathogens 11, e1005194). The metal ion composition and positioning in these X-ray structures was meticulously characterised in crystals using anomalous scattering from metal soaks combined with data collected on Fe and Mn edges as well as X-ray fluorescence experiments and by ICP-MS on solution samples. Therefore, our interpretation in the cEM structures is based on interpretation from these crystal structures. To clarify, in our revision we now say these are consistent with the crystal structures where the ion identity was assessed and have added a statement and references on page 11 that the interpretation of the density assumes Fe and Mg based on previous high-resolution crystal structures.

4) cryogenic-EM (cEM)", cryo-EM is the standard term.

While we have no problem with the term cryo-EM, cryogenic-EM is also in wide use, and we will defer to journal policy.

5) Time-point $T=10s$, is actually taken after 30 sec of incubation on the grid (4 C) and some extra time for solution application, so is it actually $\sim 60s$ after reaction initiation.

We thank the reviewer for pointing this out. In fact, the 30 s incubation was used in our study of the reconstruction of the inhibited state, and not in the time course of the live reaction. In the time course study, we blotted immediately after a 10 s wait, and I have now amended the methods to state this.

6) According to Methods section, state V was refined to C1 or C2 symmetry. Was there any difference? Did the resolution of all areas improve by imposing D2 and C2 symmetry in other structures?

To clarify, maps for all states were initially reconstructed and refined with C1 symmetry. For these maps, the active site and allosteric site were evaluated for its nucleotide and metal content. These indicated that maps had additional symmetry (C2 or D2) with similar occupancies in the symmetry related binding sites. These were subsequently refined with the higher symmetry. In all cases, the map-map FSC improved, and led to better resolution of nucleotides and metals and better atomic models.

7) It is often the case in multimeric proteins that not all active sites are similar since the substrate occupancy might vary in each pocket. The authors should analyze how pockets vary for each state

As stated in answer to point 6, occupancies in monomers appeared to be the same and occupancy high enough to produce improved maps of nucleotides and metals. We cannot exclude the possibility of additional asymmetric states with much higher resolution differences than reported in this study. However, the results are a dramatic advancement beyond the static picture from crystal structures, and we believe our interpretation of the live reaction using the methodology described in the manuscript has been thorough.

8) Overall, the authors need to validate their result with better quality maps and improve the resolution of the data.

We have carefully examined our data processing at the referee's suggestion to improve the resolution while also confirming symmetry. Some additional reprocessing has led to some improvement of maps for State-II: (C1 3.0 Å: C2 2.82 Å), State-IV: (C1 3.13 Å: C2 2.98 Å) and State-V: (C1 3.53 Å: C2 3.43 Å). We have modified the tables and our depositions accordingly.

While these modest resolution improvements are welcome, we hope we have already provided support for our atomic models and interpretations, and hope that the referee agrees, based on our re-calculation of sigma values, clarity on where ion identity is supported by high-resolution X-ray structures, and better presentation of map density in figures. Regarding the latter, these can additionally be assessed from our deposited maps and models.

II) Figures and text.

Figure 1 is difficult to understand. Depicting the tetramers as surface representations might not highlight the differences between the states; for example, comparisons between panel a (tense) and hemi-relaxed,

panel c, are completely not obvious. The authors need to show perhaps a cartoon representation overlaying the different states and zooming in on the substrates, and the allosteric sites, which are practically invisible in the figure. Are there any differences in the overall dimensions (length and width) of the tetramer between the tense and relaxed structures and between states. The figure legend is huge, needs to be summarized better.

These details regarding the active and allosteric sites have whole six-panel figures devoted to them Fig 3 and Fig 6, plus a new Figure 2 and supplementary figures now showing density, Supplementary Figure 8, and H-bond distances, Supplementary Figure 10, for nucleotides and metal ions, where resolved. The point of Figure 1 was to present the type of overview that the reviewer is suggesting and then to introduce the details later. Nevertheless, to improve the clarity and message of Figure 1 we have now made substantive changes. First (see also the request of Reviewer-1), we have moved the fraction of each state vs reaction time data from the supplementary material into this main figure. This we agree is an important point and now helps introduce the Figure 1 content. Moreover, to further address the differences in States early on in the manuscript we have edited Supplementary Movie 1 with the text ideas suggested by Reviewer-1 and presented it in the first section of the results so the reader can see the large-scale quaternary changes between states straight away. Further, we have re-worked Figure 1 to include Assembly, Steady-state and Depleted stages to show which states are present at each stage. Moreover, we have included a new Supplementary Figure 9 that indeed zooms in on the structures to show the exposure of active and allosteric sites between States. We think this further helps with clarity. In response to the reviewer's comment regarding the legend even with the inclusion of the additional new panel 1a, through adjustments to the figure, we have been able to significantly streamline it.

Are there any differences in the overall dimensions (length and width) of the tetramer between the tense and relaxed structures and between states.

There are indeed differences in dimensions between tense and relaxed tetramers. These differences are described in some detail in the results section "C-terminal lobe order-disorder transition in catalysis" and shown in the new Figure 4 that we have now moved from the supplementary material (originally Supplementary Figure 12) to the main text.

Line 123, it is hard to quantify the metal content at this resolution.

I think we can safely say that our Fe occupancy in the active site is sound. It is based on our previous crystal structure data (see response to point 3) and is very strongly liganded by the two His and 2 Asp sidechains that give the protein its name. In all the cEM maps presented in the manuscript and every SAMD1 crystal structure we have determined > 30 we see a strong density for Fe in the active site and Mg in the allosteric site backed up by previous meticulous and extensive biochemical, enzymological and structural characterisation. Additionally, these cEM data were recorded on the same protein preps, possibly even the exact same sample as the X-ray data so it is difficult to see how metal composition of the same material can be different from that measured in a crystal to that imaged by cEM. Where we do observe differences is in the Mg metal content at the active site that is highly correlated with the loss of nucleotides. If there is no nucleotide there is

accompanying loss of Mg² because it is largely only co-ordinated by the β and γ phosphates of the substrate. Mg³, is co-ordinated by both substrate and active site residues (H233 & D207) and appears to remain in the States-II and -III that interconvert with loaded State-I tetramers at steady state. However, in the depleted stage where State-V dominates it too is lost leaving only the very tight HD-bound Fe in the active site. We agree with the reviewer it would indeed be very difficult to quantify the metal content if we had no prior knowledge. However, given the enzymological, biochemical and structural data we and others have previously accumulated we feel highly confident in our assignment of metal ions in the active and allosteric sites across the states we observe.

Supplementary figure 2 doesn't show the 3-D class for the dimer.

We have now added the processing of the Dimer data to the scheme to now include the 3D class. In addition, we have now added separate exemplar streams for processing the T=600 s (State-I, -II, -III) and T=2400 s (State-IV, V) data.

Supplementary figure 3 (panels a and b) should show the resolution for each state.

FSC resolution cut-offs are now included in Figure panels a & b as well as stated in the legend.

Supplementary figure 6 shows the topology of a monomer observed in previous crystal structures, what is the point.

We feel it is necessary to have a SAMHD1 monomer and topology figure that orients the reader with where the catalytic and regulatory lobes sit with respect to the sequence and secondary structures. This is especially relevant when we discuss secondary structure elements in the text for example α 13 on Line 153, " α 13- α 13" on line 408, β 7- α 18 on Line 311 and 318. The monomer structure and topology diagram are now incorporated into an amended Supplementary Figure 5 in the revised manuscript that includes additional shading to address the reviewers point regarding topology of the different states observed by cEM.

In addition, we can see that, as suggested, it is also important to familiarise readers with residues that make interactions at the Dimer-1 and Dimer-2 interfaces. Therefore, we have prepared a new Figure, Supplementary Figure 6, that presents the SAMHD1 dimers and identifies all the key residues at the interfaces mentioned in the text. This helps satisfy the reviewer's comments below regarding Lines 149-155 and 180-184.

Authors should show the topology of the different cryoEM states and compare with the crystal structures.

The structural differences between states all relate to the degree of disorder in the lobe-linkers of the regulatory domains. Therefore, the changes to topology are small and confined. To make this clear we have introduced additional shading around the 507-545 region in the topology diagram and referred to this in the legend being the region that undergoes progressive disorder from States-I to -V.

Perhaps this depiction could allow the reader to understand where the monomer/dimer interactions take place.

The new Supplementary Figure 6 now shows the Dimer-1 and Dimer-2 interfaces and depicts all the interactions.

Lines 149 to 155, where are all these residues and interactions illustrated.

The interactions at the Dimer-1 interface that are discussed in Lines 149-155 are now described in the new Supplementary Figure 6.

Supplementary figure 7, colors are hard to differentiate.

We have now prepared an amended Supplementary Figure 7 largely to accommodate the reviewer's point below regarding fitting of Dimer-2 into the density. We have adjusted the colouring to see the density better.

Line 160. authors should show how dimer 2 does not fit in the density.

We have now included the Dimer-2 model alignment into the map in Figure 7. It is apparent that Dimer-1 fits well into the density whilst Dimer-2 does not.

Supplementary figure 8 shows the best density maps in the presence of the non-hydrolyzable analog; however, density for metals in panel e is not clear.

We have now prepared density figures without the protein backbone representation and included them in an amended Supplementary Figure 8. These show much more clearly the ion positions in the map at the allosteric and active sites. Additionally, we have included the atom and nucleotide Q scores in the figure to show that the assigned features are resolved.

Line 172, what does "modeled" mean?

The term "modelled" just refers to placement of dCTP in the active site over any other specific dNTP or generic purine or pyrimidine. For physiological relevance, the cEM data was collected on the 4 dNTP + GTP mixture that we conducted the NMR experiments on and is most close to the substrate composition in-cell. This will result in a degree of averaging regarding the identity of the base at each site. However, several studies including our own (refs 51-55 & 62) have looked at the nucleotide affinity/preference at each site using structural and enzymological methods and shown conclusively, even exquisitely (ref 52) what these preferences are. GTP is the physiological ligand for AL1, although dGTP can bind but is being turned over in the reaction. In AL2, purines are strongly preferred over pyrimidines and out of G and A, dATP has the highest affinity. In the active site, nearly all the specificity is for the triphosphate and deoxyribose groups. There is much less selection but, in a dNTP mixture dCTP and dGTP are turned over faster than TTP and much faster than dATP (see Supplementary Figure 1). We chose dCTP over dGTP as the density for the base much better accommodates the smaller pyrimidine over the two-ring purine. For further clarification, we have now expanded the text around line 72 to state that "we have placed" dCTP in the active site based on the previously identified selectivity preferences.

Lines 175-179, authors show illustrate this arrangement with a cartoon.

We have now included the symmetry state in the new Figure 1 to illustrate that State-I is a D2 structure with 4 equivalent active and allosteric sites and that State-II to V are C2 structures that contain 2 pairs of equivalent active sites two on the tense (B and D side and two on the relaxed (A and C side). In addition, we now reference the Movie that very clearly shows the transition between states in the first section of the Results and clearly assigned each state to the phases of the reaction where it is prevalent. See also response to “*figures and text*” point 1.

Lines 180-184, not clear where these residues and interactions are located, should be shown.

The interactions at the Dimer-2 interface that are discussed in lines 175-179 are now shown and described in the new Supplementary Figure 6.

Lines 183-185, authors should overlay cartoon representations to give the reader a sense of the differences between the tense, relaxed and semi-relaxed structures.

We have now referenced the movie at the start of the results section which clearly shows the tense and relaxed sides of the C2 tetramers. We have now also referenced the movie in the text on line 185 and on line 303.

Supplementary figure 9, is one of the most relevant figures of the paper, since it validates the results of Figure 2. Panel a shows density at the active site for the substrate and the metals. This should be made clear with a better depiction of the maps. As mentioned before, 1.5 sigma is way too low. It ought to be shown that this density is not noise. Can they base type be identified? The active site of panel b does not illustrate the product. The density is not clear at all. It is hard to see how the metals in panels c-e would stay in place; maps illustrating the coordinating residues and distances should be included to validate the results.

We have now produced much better-quality density figures and removed the cartoon of the protein backbone. The figure now shows very clearly the fitting of nucleotides and metals into density and so we have now included it as part of a new main text Figure 2. We thank the reviewer for the suggestion.

Bases can be identified in allosteric sites based on prior knowledge from our and other high resolution crystal structures, our own high-resolution inhibitor and State-I cEM structures, a knowledge of what was included in the experimental set of up and the site base preferences established through enzymological studies. Therefore, we have assigned GTP and dATP to the allosteric site and dCTP to the active site as these are the most preferred through structural enzymological and chemical constraints. See also response to “*what does modeled mean*”. Additionally, we have produced a new Supplementary Figure 10 to illustrate metal-ligand interactions and side chain H-bonding to nucleotides.

Figure 2 needs validation with maps. H-bond distances and coordination distances of metals should be indicated to validate lines 213-216.

In the production of the new Figure 2 we now show more clearly density for metal ions and nucleotides in the active and allosteric sites. In addition, we have prepared a new supplementary Figure 10, that displays the H-bond distances and ligand co-ordination of the active site metals in State-I and State-III. This figure now also contains the State-I – State-III overlap as a new panel Figure 10c.

Lines 216-225 need map validation; the resolution is too low to make these conclusions.

In the production of the new Figure 2 we now show more clearly density for metal ions and nucleotides in the active and allosteric sites. For State-II, density for the contents of the relaxed A and C active sites is weaker than in tense B and D, which might be expected for mobile reaction products. Whereas there is clearly no density other than the ions in State-III, -IV and V. Moreover, we see a clear discontinuity that locates between a modelled α -phosphate of a triphosphate and the 5'-hydroxyl of the sugar of a deoxycytidine nucleoside. This gives us the confidence to assign it as a product complex. Nevertheless, on lines 218-220 we have now acknowledged that the density for contents of the State-II A and C active sites is weak and that there is a degree of disorder associated with it.

Line 226, the authors should point to supplementary figure 10H (also line 235), which depicts these conformational changes.

We have now referred in the text to the new Supplementary Figure 10 that includes the overlay as Figure 10c showing the side chain conformation shift between State-I and -III.

Lines 228-231 is this true for each monomer?

Maps are symmetrised so by definition all 4 active sites have the same content and configuration in State-I (D2). The State-II and State-III maps are C2 so in State-II Active sites A and C are identical and have product, B and D have substrate. In State-III active sites A and C are empty B and D have substrate. We very carefully analysed all maps in C1 to look for ligand differences before then assigning higher symmetry. Maps were then refined in higher symmetry and in all cases map-map FSC improved, density for ligands was better resolved and better fit of atomic models. See also response to the "EM data" points 6,7 and 8.

Figure 2 should include overlay of residues to illustrate conformational changes mentioned in lines 237-239.
We have now referred to Supplementary Figure 10c that illustrates the side chain conformational changes.

Figure 3 should show how the catalytic sites are exposed during the order/disorder transition in catalysis (line 277).

We have replaced the original Figure 3, now Figure 4, with the original much more detailed Supplementary Figure 12. We have also created a new supplementary Figure 9 that now shows more clearly exposure of the active sites, see response below.

Figure 4. There is no good color contrast for panels a-c. Panels d-f should show the substrate and how the locked-engage-release conformers relate to substrate binding, catalysis and release.

Panels a-c indeed show the exposure of the active sites but were perhaps a little small and Reviewer-1 thought the figure superfluous. The main purpose was to orient the reader in the structure with respect to the lower panels that contain the H-bonding details of the C-terminal lobe – linker interactions. We have now edited the Figure to combine panels a-d, b-e and c-f and box the regions of interest in the upper panels. The legend has

also been amended to connect the relationship more fully between each upper and lower panel. See also response to Reviewer-1. In addition, we have now prepared a further supplementary Figure 9 that also now shows more clearly the exposure of active sites during the catalytic cycle, see response above.

Discussion should include a schematic of the proposed cycle, from assembly to substrate binding, catalysis, and release.

We believe this point is addressed by our modifications in the new Figure 1 and the improved Supplementary Movie that we have included in the Revision.

Reviewer-3

Acton et al, Platform-directed allostery and quaternary structure dynamics of SAMHD1 catalysis. SAMHD1 is deoxynucleotide triphosphate hydrolase and a critical regulator of cellular dNTP homeostasis. The authors have used time-resolved cryoEM to visualize the assembly and allostery of SAMHD1. They were able to observe how conformational changes in the tetramer drive the catalytic cycle. Specifically, they identified 5 different conformational states and propose that opening and closing of the active sites drives enzyme catalysis. This is an excellent manuscript thoroughly describing the detailed mechanism of SAMHD1 that will be of relevance to those interested in nucleotide metabolism, cell cycle regulation or multi-subunit enzyme regulation. The figures are clear and well-presented and the conclusions drawn from the data are logical. The structures seem to be well determined and the additional data solid.

We thank the Reviewer-3 their appreciation of the manuscript.

I only have a couple of very minor comments for the authors' consideration:

In the first sentence of the introduction, the authors state that SAMHD1 is a dNTPase and provide a reference. I think it would be more appropriate to have the original description of the dNTPase activity added to the references (including the authors own paper) PMID:22056990 and PMID: 22069334

We have now included the original references to SAMHD1 dNTPase activity.

Pg 5 Line 78, The authors state that phosphorylation regulates dNTPase activity. This idea is fairly controversial, and there is significant data that suggests otherwise. The authors may want to modify or qualify the statement.

We agree the relationship between SAMHD1 phosphorylation, dNTPase activity and HIV-1 restriction is controversial. I have now added two additional sentences in the introduction to clarify that there are opposing views of the exact role of SAMHD1 phosphorylation.

It is curious that the SAM domain does not have any density in any of the structures. Just curious if the authors have thoughts about if or how the SAM domain may contribute to the structural changes or conformational states observed.

The SAM domain in our hands has remained extremely elusive. We have never observed it in crystal structures of full length SAMHD1 constructs. In this study we see some very low-resolution density in the dimer map that we mention (page 8). In previous studies we and others have shown that in human SAMHD1, its presence has little effect on tetramer assembly and actually slightly reduces k_{cat} with respect to a truncated construct. By contrast, in murine SAMHD1, the SAM domain is required for complete tetramer assembly and dNTPase activity by acting to cap off the AL1-AL2 sties and stabilise the

active form of the enzyme. As we don't see the SAM domains in any of our tetramer states it is difficult to speculate on its contribution. Potentially it could act to control loss of nucleotides from the allosteric sites as the low-resolution density we do observe in the dimers is close in space to AL1-AL2 but that would be a hand-wavy explanation at best.

REVIEWERS' COMMENTS

Reviewer #2 (Remarks to the Author):

The authors now present a much improved version that is suitable for publication.

Some remarks:

1) Not sure if discussion about the pair distribution function and the radius of gyration is really reinforcing or distracting in the paper.

2)Line 178:

"Although data was collected from a mixed dNTP reaction based upon the previously identified selectivity preferences^{51-55,62}, in our models we have placed dCTP at the active site and dATP at AL2 (Fig 2a)."

But on Line 115:

"the reaction approximates to steady-state conditions with some substrate preference in the rank order of k_{cat} with $dGTP > dCTP > TTP > dATP$." With $dGTP$ (0.249 s⁻¹) and $dCTP$ (0.201 s⁻¹)" the authors should clearly state that they have a mixture, impossible to determine which dNTP is really on the active site, and they are just modeling since at this resolution it is not possible to select the right one.

Response to reviewers: manuscript NCOMMS-23-53244A Acton et al 2024

Reviewer #2 (Remarks to the Author):

The authors now present a much improved version that is suitable for publication.

We thank Reviewer-2 for the continued assessment of the manuscript and final comments.

Some remarks:

1) Not sure if discussion about the pair distribution function and the radius of gyration is really reinforcing or distracting in the paper.

The C-terminal lobe-linker order-disorder transition is a major finding in the paper. It relates quaternary changes in the regulatory domains directly to the release of products and reloading of substrates in the active sites through the catalytic cycle. The movie we present gives a good visualisation of the process. However, R_g and D_{max} are a simple metric that reports on the compaction and spacing of the regulatory domains and provides a description of the domain movements from compact, State-I ($R_g = 28.6 \text{ \AA}$) to relaxed, State-III (32.6 \AA) and depleted State-V ($R_g = 34.6 \text{ \AA}$). Further, the pair-distribution function reports this disorder as an increase in bi-modality going from State-I to State-V. We think this simplified way of expressing, what are quite complex re-arrangements, is an excellent way of describing the system and would want to maintain it in the manuscript.

2)Line 178:

“Although data was collected from a mixed dNTP reaction based upon the previously identified selectivity preferences^{51-55,62}, in our models we have placed dCTP at the active site and dATP at AL2 (Fig 2a).” But on Line 115: “the reaction approximates to steady-state conditions with some substrate preference in the rank order of k_{cat} with $dGTP > dCTP > TTP > dATP$.” With $dGTP (0.249 \text{ s}^{-1})$ and $dCTP (0.201 \text{ s}^{-1})$ ” the authors should clearly state that they have a mixture, impossible to determine which dNTP is really on the active site, and they are just modeling since at this resolution it is not possible to select the right one.

On Line 178, We have now amended the sentence to clarify the mixed occupancy at active sites. We now state “As these data were collected from a four-dNTP reaction, based on previously identified selectivity preferences^{51-55,62}, in our models we have placed dCTP at the active site and dATP at AL2 (Fig. 2a) although it is likely the active sites have mixed dNTP-base occupancy”.